# APC/C$^{CDH1}$ synchronizes ribose-5-phosphate levels and DNA synthesis to cell cycle progression

Yang Li[1,2,7], Cui-Fang Yao[1,7], Fu-Jiang Xu [1,3], Yuan-Yuan Qu[3], Jia-Tao Li[1], Yan Lin[1,2], Zhong-Lian Cao[4], Peng-Cheng Lin[5], Wei Xu[1,2,6], Shi-Min Zhao[1,2,6] & Jian-Yuan Zhao[1,2,6]

Accumulation of nucleotide building blocks prior to and during S phase facilitates DNA duplication. Herein, we find that the anaphase-promoting complex/cyclosome (APC/C) synchronizes ribose-5-phosphate levels and DNA synthesis during the cell cycle. In late $G_1$ and S phases, transketolase-like 1 (TKTL1) is overexpressed and forms stable TKTL1-transketolase heterodimers that accumulate ribose-5-phosphate. This accumulation occurs by asymmetric production of ribose-5-phosphate from the non-oxidative pentose phosphate pathway and prevention of ribose-5-phosphate removal by depleting transketolase homodimers. In the $G_2$ and M phases after DNA synthesis, expression of the APC/C adaptor CDH1 allows APC/C$^{CDH1}$ to degrade D-box-containing TKTL1, abrogating ribose-5-phosphate accumulation by TKTL1. TKTL1-overexpressing cancer cells exhibit elevated ribose-5-phosphate levels. The low CDH1 or high TKTL1-induced accumulation of ribose-5-phosphate facilitates nucleotide and DNA synthesis as well as cell cycle progression in a ribose-5-phosphate-saturable manner. Here we reveal that the cell cycle control machinery regulates DNA synthesis by mediating ribose-5-phosphate sufficiency.

[1] Obstetrics & Gynecology Hospital of Fudan University, State Key Laboratory of Genetic Engineering, School of Life Sciences and Institutes of Biomedical Sciences, Fudan University, Shanghai 200438, P.R. China. [2] Key Laboratory of Reproduction Regulation of NPFPC and Collaborative Innovation Center for Genetics and Development, Shanghai Key Laboratory of Female Reproductive Endocrine Related Diseases, Fudan University, Shanghai 200438, P.R. China. [3] Fudan University Shanghai Cancer Center, Fudan University, Shanghai 200438, P.R. China. [4] School of Pharmacy, Fudan University, Shanghai 200438, P.R. China. [5] Key Laboratory for Tibet Plateau Phytochemistry of Qinghai Province, College of Pharmacy, Qinghai University for Nationalities, Xining 810007, P. R. China. [6] Collaborative Innovation Center for Biotherapy, West China Hospital, Sichuan University, Chengdu 610041, P.R. China. [7] These authors contributed equally: Yang Li, Cui-Fang Yao. Correspondence and requests for materials should be addressed to S.-M.Z. (email: zhaosm@fudan.edu.cn) or to J.-Y.Z. (email: zhaojy@fudan.edu.cn)

Cell cycle progression is timely and precisely regulated by the anaphase-promoting complex or cyclosome (APC/C) and the Skp1/Cul1/F-box protein (SCF) E3 ubiquitin ligase complexes, which mediate the degradation of cell cycle-regulating protein substrates[1,2]. SCF controls the transition between $G_1$/S and $G_2$/M phases, depending on the levels and activities of over 70 adaptor F-box proteins[3,4]. APC/C complexes with the substrate-recruiting CDC20 (APC/C$^{CDC20}$) to orchestrate cell exit from mitosis and tethers with another substrate-recruiting protein, CDH1 (APC/C$^{CDH1}$), to establish a stable $G_1$ phase[2]. During mitotic exit, APC/C$^{CDH1}$ ubiquitinates CDC20, polo-like kinase 1, and aurora kinases. APC/C$^{CDH1}$ is then inactivated by a combination of CDH1 phosphorylation, ubiquitylation that degrades CDH1, and binding to the APC/C inhibitor—early mitotic inhibitor 1 (EMI1)[5]. Besides being detected in the nucleus, F-box proteins are also distributed in the cytoplasm[6,7], while the APC/C localizes to centrosomes, microtubules, chromosomes, and kinetochores during mitosis[8,9], indicating that APC/C and SCF may target substrates other than cell cycle-regulating proteins and regulate other cellular processes. Indeed, APC/C$^{CDH1}$ was found to degrade 6-phosphofructo-2-kinase/fructose-2,6-bisphosphatase isoform 3 (PFKFB3)[10].

Timely and quantitatively precise synthesis of cellular molecules, such as DNA, RNA, and proteins, is a prerequisite for well-orchestrated cell cycle progression. However, the underlying mechanisms of cell cycle-dependent regulation for these anabolic processes remain unclear. For example, how ribose-5-phosphate (R5P), an intermediary metabolite of the pentose phosphate pathway (PPP), required for both de novo and salvage synthesis of nucleosides and, consequently, for doubling of DNA and RNA synthesis, is regulated during cell cycle progression remains uncertain[11–13] The current paradigm is that R5P is pulled in to the nucleotide and DNA synthetic pathways from the PPP and that overproduction of nucleotides is prevented by feedback inhibition. For example, purine-5′-nucleotide (AMP and GMP) feedback inhibits X-chromosome located ribose-phosphate pyrophosphokinase 1 (PRPS1), ribose-phosphate pyrophosphokinase 2 (PRPS2), and chromosome 7p21.1-located ribose-phosphate pyrophosphokinase 1-like 1 (PRPS1L1), preventing these isozymes from converting R5P into 5-phosphoribosyl-1-pyrophosphate (PRPP)[13–15]. Adenosine and guanosine nucleotide feedback inhibits PRPP amidotransferase, while pyrimidine feedback inhibits the trifunctional carbamoyl-phosphate synthetase-2, aspartate transcarbamoylase, and dihydroorotase (CAD)[16]. Although the Pulled-In theory is supported by several observations, including that PRPP-producing pyrophosphokinase (PRPS)[13] is activated in rapidly-growing cells[17,18] and that the concentration of nucleotides increases from late $G_1$ to S phase and then decreases after completion of DNA duplication[19], this model does not explain whether and how R5P sufficiency in PPP is attained from late $G_1$ to S phase. In fact, proactive R5P-accumulating mechanisms may also induce consequences such as PRPS activation and nucleotide increases from late $G_1$ to S phase. Evidence is also accumulating that supports R5P sufficiency in PPP as a determining factor of DNA synthesis. For example, enhanced PPP by either p53 or TAp73 led to accelerated DNA synthesis[20,21]. Moreover, increased intracellular concentrations of R5P were observed when cells advanced toward the late $G_1$ and early S phase[22], an argument against the Pulled-In model, which predicts decreased intracellular concentrations of R5P during late $G_1$ and early S phases due to increased consumption. Furthermore, it has been found that the majority of R5P incorporated into de novo and salvage purine synthesis in S phase originates from non-oxidative PPP[23]. This finding implies that reprogrammed R5P metabolism occurs before DNA synthesis, since pulling in R5P would not preferentially incorporate R5P from non-oxidative PPP.

R5P is synthesized from glucose-6-phosphate-derived ribulose-5-phosphate by ribulose-5-phosphate isomerase from the oxidative branch of the PPP, and from glyceraldehyde-3-phosphate (G3P) by transketolase (TKT) of the non-oxidative branch of PPP[24]. Both routes are directly linked to the glycolytic pathway. TKT plays dual roles in controlling R5P levels; it removes R5P by converting it to G3P and sedoheptulose-7-phosphate (S7P) when xylulose-5-phosphate (X5P) and R5P are used as substrates and can catalyze the formation of R5P from G3P and S7P. The bidirectional activities of TKT make it a possible key R5P-regulating enzyme that determines R5P levels by changing catalytic directions and altering substrate specificities.

The human genome encodes two proteins closely related to TKT, transketolase-like protein 1 and 2 (TKTL1 and TKTL2)[25]. TKTL1 shares 61% amino acid sequence identity with TKT. However, the TKT activity of TKTL1 is yet to be confirmed, especially through in vitro assays, although a correlation between TKTL1 and total cellular TKT enzymatic activity was observed in cells[26–28]. Compared with TKT, the most dissimilar region of TKTL1 was a 38-amino acid deletion in the N-terminus. Structural studies found that TKT proteins harboring this deletion lack TKT activity, suggesting that TKTL1 lacks TKT activity[29,30], while a biochemical study detected certain levels of TKT activity for TKTL1[31]. Nevertheless, evidence suggests a role for TKTL1 in proliferation or cell cycle regulation. Indeed, TKTL1 is overexpressed in various cancers and is correlated with poor prognosis in colon, urothelial, gastric, and lung cancers as well as in ocular adnexa carcinomas, rectum carcinomas, and laryngeal squamous cell carcinomas[32–41]. Increased TKTL1 levels also correlate with esophageal squamous cell carcinoma metastasis and increased resistance against cisplatin chemotherapy in nasopharyngeal carcinomas[42,43]. Moreover, TKTL1 overexpression promotes cell proliferation and enhanced tumor growth[26]; in contrast, TKTL1 downregulation attenuates the proliferation of various types of cancer cells[44–46]. Notably, it has been suggested that TKTL1 regulates R5P levels[37].

In the present study, we find that TKTL1 is controlled by APC/C$^{CDH1}$ in a cell cycle-dependent manner and that TKTL1 expression is associated with R5P regulation. We investigate how APC/C$^{CDH1}$ regulates R5P levels and DNA synthesis by targeting TKTL1.

## Results

**TKTL1 levels vary throughout the cell cycle.** When human cervical cancer HeLa cells were synchronized to $G_1$/S phase by double thymidine (Supplementary Fig. 1a; 0h), endogenous levels of TKTL1, but not TKT and TKTL2, increased. Meanwhile, endogenous levels of TKTL1, but not TKT and TKTL2, decreased in HeLa cells synchronized to $G_2$/M phase by RO3306 (Supplementary Fig. 1b, 0h) (Fig. 1a). These observations suggest that TKTL1 protein levels exhibit cell cycle-dependent regulation. To test this hypothesis, we released the arrested cells for progression into subsequent phases of the cell cycle. After double thymidine-synchronized $G_1$/S phase HeLa cells were released, TKTL1 protein levels remained elevated during S phase, decreased when cells progressed into $G_2$/M phase, and bounced back after cells re-entered $G_1$ phase (Fig. 1b and Supplementary Fig. 1a; 0–11h). TKTL1 protein levels also increased after RO3306-synchronized $G_2$/M phase HeLa cells entered $G_1$ phase (Fig. 1c and Supplementary Fig. 1b; 0–8h). The TKTL1 levels were inversely correlated with cell cycle-regulating CDH1 levels, while the levels of other R5P metabolism-associated enzymes—TKT, ribose 5-phosphate isomerase A (RPIA), and the potential transketolase TKTL2—did not change as the cell cycle progressed (Fig. 1b, c). These results indicate that TKTL1 levels are regulated during cell cycle progression.

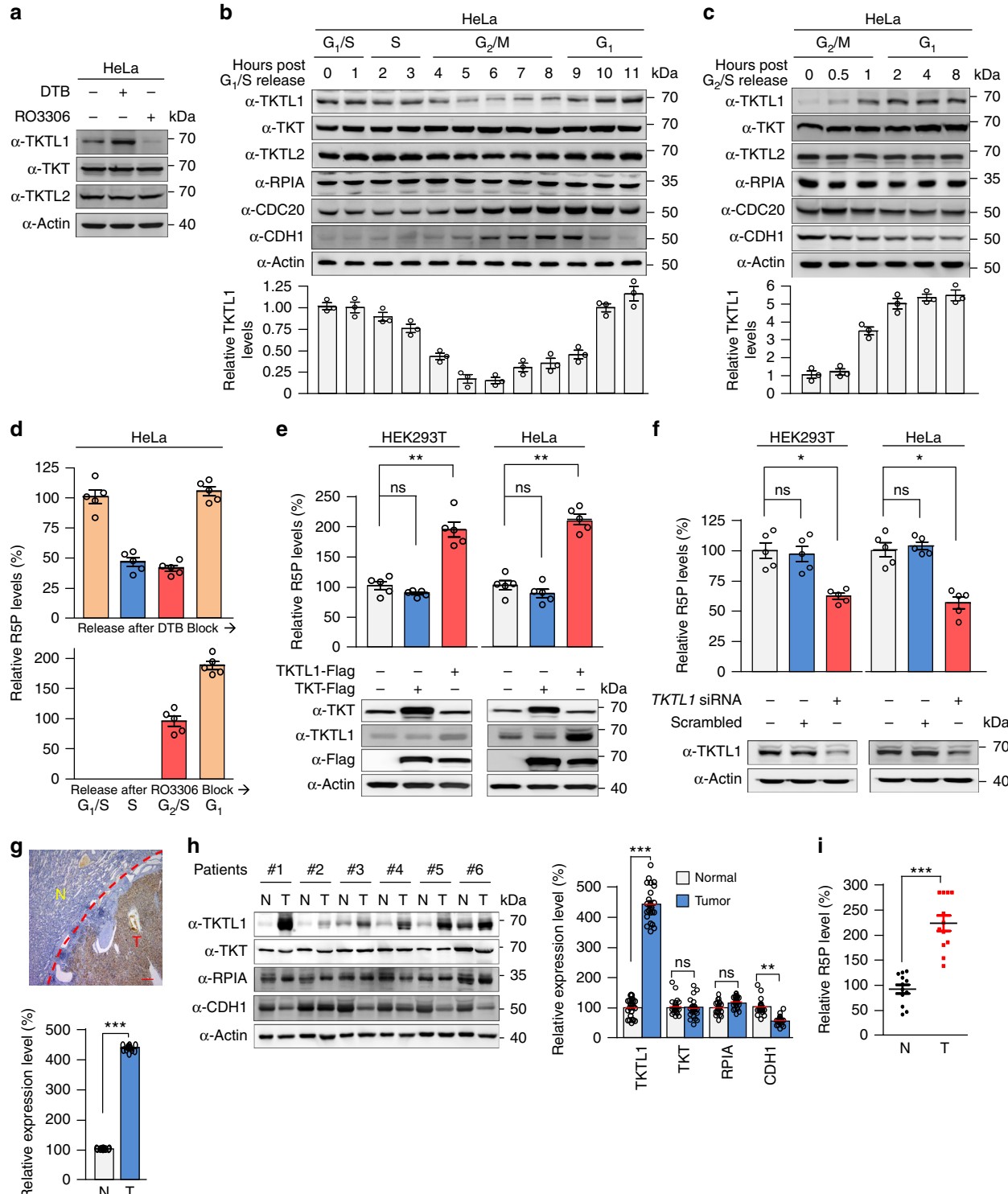

**TKTL1 upregulates R5P levels**. Analysis of R5P levels in HeLa cells at different cell cycle phases, employing liquid chromatography–mass spectrometry (LC-MS), revealed that R5P levels peaked during the $G_1/S$ phase, decreased after entering S phase, and remained low during the $G_2/M$ phases (Fig. 1d), indicating that R5P levels increase during $G_1$ phase and are substantially consumed in S phase. The positive correlation between TKTL1 levels and R5P levels (Fig. 1b–d) during the cell cycle suggests that TKTL1 may be a positive regulator of R5P. To test this possibility, we determined R5P levels in TKTL1-overexpressing and knockdown cells. While TKT

overexpression caused negligible changes in R5P levels, TKTL1 overexpression resulted in a nearly twofold elevation in cellular R5P levels in both human embryonic kidney HEK293T and HeLa cells (Fig. 1e). Conversely, TKTL1 knockdown led to a 35% and 40% decrease in cellular R5P levels in HEK293T and HeLa cells, respectively (Fig. 1f). These results support the hypothesis that TKTL1 positively regulates R5P levels in cultured cells. Since TKTL1 is overexpressed in many cancer types[32–37], including clear cell renal cell carcinoma (ccRCC) tissues (Figs. 1g, h and Supplementary Fig. 2a, b), we compared R5P levels between ccRCC tissues and corresponding adjacent non-cancer tissues.

**Fig. 1** Cell cycle-coupled TKTL1 expression regulates R5P levels. **a** Protein levels of TKTL1, TKT, and TKTL2 were determined in double thymidine-synchronized $G_1/S$ and RO3306-synchronized $G_2/M$ HeLa cells. Synchronizing effects were detected by flow cytometry (see Supplementary Fig. 1). **b**, **c** Protein levels of TKTL1, TKT, TKTL2, RPIA, CDC20, and CDH1 were determined at different time points after HeLa cells were released from (**b**) double thymidine and (**c**) RO3306 synchronization. Quantitative results of TKTL1 are shown below. Values are means ± SEM of three independent experiments. The cell cycle phases of indicated time points were confirmed by flow cytometry (see Supplementary Fig. 1). **d** R5P levels in HeLa cells at different cell cycle phases were measured after cells were released from double thymidine (upper panel) and RO3306 (lower panel) synchronization. Cell phases were confirmed by flow cytometry sorting. Data are presented by means ± SEM of five independent experiments. **e** R5P levels in both HEK293T and HeLa cells as well as TKT or TKTL1-overexpressing HEK293T and HeLa cells were determined. Data are shown as average of five independent experiments, presented as means ± SEM, two-tailed Students' $t$ test, **$p < 0.01$, ns not significant. **f** R5P levels in both HEK293T and HeLa cells as well as TKTL1-knockdown HEK293T and HeLa cells were determined. Values are shown as average of five independent experiments, presented as means ± SEM, two-tailed Students' $t$ test, *$p < 0.05$, ns not significant vs. control group. **g** Overexpression of TKTL1 in ccRCC. Representative immunohistochemical staining (IHC) and quantitative results of 12 samples are shown. T, tumor tissue; N, adjacent non-cancer tissue. Scale bars: 200 μm. Data are presented as means ± SEM and two-tailed Students' $t$ test was used, ***$p < 0.001$. **h** Expression levels of TKTL1, TKT, RPIA, and CDH1 in ccRCC. Protein levels of ccRCC tumors and matched adjacent non-cancer tissues were analyzed by western blotting (left). Quantitative results ($n = 24$ pairs of ccRCC tumors tissues and adjacent tissues; right) are shown as means ± SEM, Student's $t$ test, ***$p < 0.001$, **$p < 0.01$, *$p < 0.05$, ns not significant. **i** Average R5P concentrations were determined for 24 paired ccRCC tumors and their matched non-cancer tissues. Data are shown as means ± SEM, Student's $t$ test, ***$p < 0.001$. Full-length blots are presented in Supplementary Fig. 10

R5P levels in ccRCC tissues were on average more than 100% higher than those in non-cancer tissues in the 24 samples tested (Fig. 1i). These observations, together with the finding that levels of the R5P-relevant metabolic enzymes TKT and RPIA did not differ between cancer and non-cancer tissues (Fig. 1h), support the hypothesis that TKTL1 positively regulates R5P levels in vivo.

**APC/C^CDH1 mediates TKTL1 proteasomal degradation**. *TKTL1* mRNA levels did not fluctuate during cell cycle progression (Supplementary Fig. 3a and 3b), excluding the possibility that TKTL1 levels are regulated at the transcriptional level. Treating HeLa cells with cycloheximide, a protein translation inhibitor, did not prevent the degradation of TKTL1 (Fig. 2a). Moreover, treatment with the proteasome inhibitor MG132 elevated cellular TKTL1 levels (Fig. 2b) and increased ubiquitination levels of ectopically expressed TKTL1 (Fig. 2c) in HeLa cells, indicating that TKTL1 levels are regulated by the ubiquitin proteasome pathway.

A CDC20 and CDH1-recognizing destruction box (D-box)[47] is present in TKTL1 but not in TKT (Fig. 2d). We therefore tested whether TKTL1 is a substrate of APC/C, which employs CDH1 and CDC20 as adaptors. In cultured cells, exogenous-tagged TKTL1 interacted with tagged CDH1 (Fig. 2e), but not with CDC20 (Supplementary Fig. 4a). Affinity purified endogenous TKTL1 was found co-purified with endogenous CDH1 from ccRCC lysates (Fig. 2f), confirming that TKTL1 and CDH1 interact with each other in vivo. Overexpressing CDH1, but not CDC20, in HeLa cells reduced endogenous TKTL1 levels (Fig. 2g). Enhanced CDH1 expression led to decreased TKTL1 protein stability (Supplementary Fig. 5a), while CDH1 knockdown increased it (Supplementary Fig. 5b), confirming that CDH1 regulates TKTL1 stability. Moreover, CDH1 overexpression in HeLa cells enhanced ubiquitination levels (Fig. 2h), whereas CDH1 knockdown by small interfering (si) RNA decreased ubiquitination levels (Fig. 2i) of ectopically expressed TKTL1 and stabilized endogenous TKTL1 (Fig. 2j). Through an in vitro ubiquitination assay, we validated APC/C^CDH1 ubiquitination of TKTL1 (Fig. 2k). Furthermore, we found that removal of the D-box from TKTL1 by simultaneously switching Arg[18] and Leu[21] to alanine (ΔD-box) weakened TKTL1^ΔD-box interaction with CDH1 (Fig. 2l), abrogated the ability of CDH1 to ubiquitinate TKTL1^ΔD-box (Fig. 2m), and rendered stable ectopically expressed TKTL1^ΔD-box levels unresponsive to CDH1 overexpression (Fig. 2n). Endogenous mutant TKTL1 levels in Crispr/Cas9-mediated TKTL1 D-box mutant knock-in cells (TKTL1^ΔD-box-knockin) did not respond to CDH1 overexpression (Fig. 2o)

and did not change throughout the cell cycle (Supplementary Fig. 5, c-e). Moreover, TKTL1 was most heavily ubiquitinated when CDH1 was highly expressed during $G_2/M$ phase (Fig. 2p). Collectively, these findings confirm that CDH1 targets and degrades TKTL1.

Although exogenous TKTL1 was found to interact with the SCF adaptors SKP2 and WD-40 domain protein 7 (FBW7; Supplementary Fig. 4b and 4c), these interactions were not observed in ccRCC tissues (Supplementary Fig. 4d). Moreover, overexpression of either SKP2 or FBW7 did not affect cellular TKTL1 levels (Supplementary Fig. 4e), indicating that SCF complexes are unlikely to be upstream regulators of TKTL1 and that TKTL1 is solely regulated by APC/C^CDH1.

**TKTL1 depletes TKT by forming a stable heterodimer with TKT**. Tandem affinity purification (TAP) analysis employing TKTL1 as bait constantly identified TKT as a binding protein of TKTL1 and TAP analysis employing TKT as bait routinely identified TKTL1 as a binding partner of TKT in HEK293T cells (Fig. 3a). These results suggest that TKTL1 forms a protein complex with TKT. Pull-down assays of purified recombinant TKTL1 and TKT (Fig. 3b) supported their direct interaction in vitro and affinity purified endogenous TKTL1 co-purified endogenous TKT from ccRCC lysates (Fig. 3c), indicating that TKT and TKTL1 also interact with each other in vivo. Moreover, surface plasmon resonance (SPR) revealed that the dissociation constant ($K_D$) of the binary TKTL1-TKT complex was 0.146 μmol/L, almost identical to the $K_D$ of TKT-TKT homodimers, which was 0.118 μmol/L (Fig. 3d); this shows that TKTL1 has as strong ability to form TKTL1-TKT binary complexes and therefore to deplete TKT dimers in cells.

To elucidate the nature of the binary TKTL1-TKT complex, we purified recombinant TKTL1-TKT complexes from *Escherichia coli* (*E. coli*) by employing a two-tags purification system as previously described[48]. Size-exclusion chromatography eluted the TKTL1-TKT complex immediately after the molecular marker MSH6 (152 kDa), which was earlier than that for the purified recombinant TKTL1 and similar to that of the purified recombinant TKT—a known homodimer[29] (Fig. 3e). This elution pattern suggests that TKTL1 forms a stable heterodimer with TKT and that TKTL1 also exists as a monomer. The dimeric TKT, TKTL1-TKT, and monomeric TKTL1 were further confirmed by native gel electrophoresis (Fig. 3f), supporting previous ratiocination that TKTL1 and TKT can form active heterodimers and play a role in the regulation of TKT activity[25]. Moreover, we further confirmed the TKTL1-TKT heterodimer by identifying binding

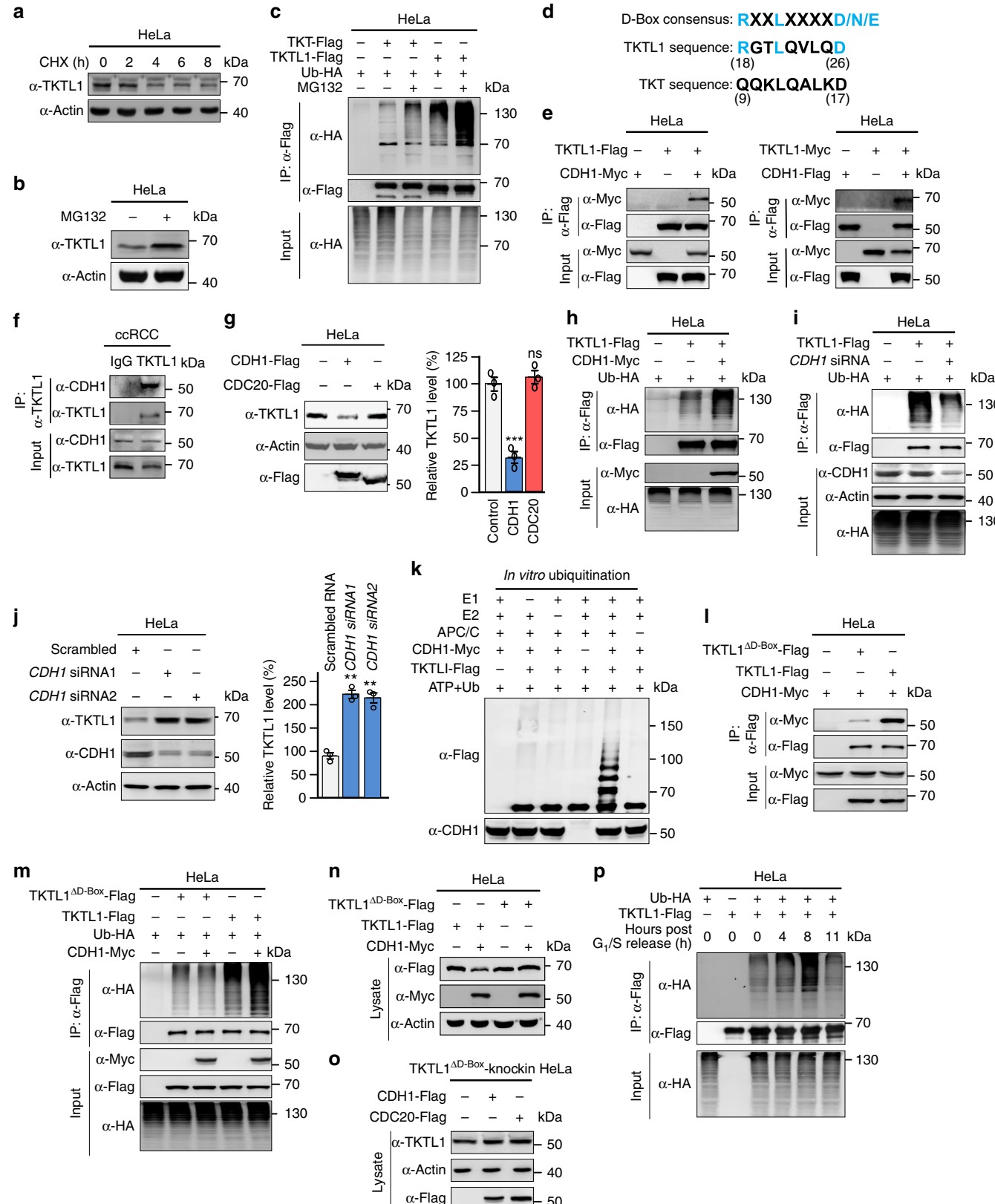

regions in TKTL1 and TKT for heterodimer formation. Truncated TKTL1 and TKT were tested for their ability to form heterodimers; when its N-terminus (a.a. 1–118) and C-terminus (a.a. 448–596) were simultaneously removed, TKTL1 lost its binding ability to TKT (Fig. 3g). Similarly, TKT lost binding ability to TKTL1 only when both the N-terminus (a.a. 1–147) and

C-terminus (a.a. 474–623) of TKT were simultaneously removed (Fig. 3h). We also generated truncated TKTL1 and TKT lacking the central portion and found that the TKTL1 termini could still interact with TKT and vice versa (Fig. 3i, j). Overall, the results reveal that TKTL1 and TKT form heterodimers via interactions at their N-terminus and C-terminus regions.

**Fig. 2** APC/C$^{CDH1}$ controls TKTL1 proteasomal degradation. **a** TKTL1 levels in HeLa cells were determined at different time points after protein synthesis was blocked by cycloheximide. **b** TKTL1 levels were determined in HeLa cells cultured with or without the proteasome inhibitor MG132. **c** TKTL1 and TKT ubiquitination. **d** The TKTL1 sequence matching the D-box consensus sequence and the TKT sequence corresponding to the TKTL1 D-box sequence are shown. **e** Co-immunoprecipitation of TKTL1-FLAG and CDH1-Myc co-expressed in HeLa cells. **f** Affinity purified TKTL1 from lysates of ccRCC tissue was probed for CDH1 to detect the in vivo interaction of TKTL1 and CDH1. **g** Endogenous TKTL1 levels were measured in HeLa cells and HeLa overexpressing CDH1 or CDC20 ($n = 3$ biologically independent samples). Quantitation of western blots is shown on the right. Error bars, mean ± SEM, Student's $t$ test. ***$p < 0.001$, ns not significant. **h** FLAG-TKTL1, Ub-HA, and CDH1-Myc were co-expressed in HeLa cells and then TKTL1 was purified by immunoprecipitation. Ubiquitination levels of TKTL1 were determined by anti-HA antibody. **i** TKTL1-FLAG, Ub-HA, and CDH1 siRNA were co-transfected in HeLa cells. Ubiquitination levels of immunoprecipitated TKTL1 were then detected. **j** Two siRNAs targeting different CDH1 regions were used. Representative western blot and mean values ($n = 3$ biologically independent samples) of quantitative western blotting results are shown. Mean values with SEM are reported, Student's $t$ test. **$p < 0.01$, ns not significant. **k** A commercial ubiquitination system was employed to detect ubiquitination of purified TKTL1 by APC/C$^{CDH1}$. **l** Amounts of CDH1 co-immunoprecipitated with TKTL1 and TKTL1$^{\Delta D\text{-box}}$ were compared when they were co-expressed in HeLa cells at comparable levels. **m** Ubiquitination levels of TKTL1 and TKTL1$^{\Delta D\text{-box}}$ were detected after they were expressed alone or co-expressed with CDH1 in HeLa cells. **n** TKTL1 and TKTL1$^{\Delta D\text{-box}}$ levels in HeLa cells were detected in the presence or absence of CDH1 co-expression in cells. **o** Endogenous TKTL1 levels were determined in TKTL1$^{\Delta D\text{-box}}$-knockin HeLa cells and TKTL1$^{\Delta D\text{-box}}$-knockin HeLa overexpressing CDH1 or CDC20. **p** Ubiquitination levels of affinity purified TKTL1-FLAG at different cell cycle phases were determined. Full-length blots are presented in Supplementary Fig. 10

### The TKTL1-TKT possesses asymmetric TKT activities.

Next, we tested the forward and reverse TKT activities (Fig. 4a) of the TKT homodimer, TKTL1 monomer, and TKTL1-TKT heterodimer—three species that should co-exist in cells. Recombinant TKT homodimers exhibited all four TKT activities, while recombinant TKTL1 virtually lacked all of them (Fig. 4b). Meanwhile, the TKTL1-TKT heterodimer exhibited reduced reaction 1 and 2 activities but elevated reaction 3 and 4 activities (Fig. 4c). Asymmetric TKT activities of the TKTL1-TKT heterodimer revealed that R5P removal (reaction 1) was impaired, whereas R5P production from non-oxidative PPP (reaction 3) was facilitated during TKTL1-TKT heterodimer formation in cells. The mechanistic basis for the asymmetric TKT activity of TKTL1-TKT heterodimers can be—at least partially—attributed to the fact that TKTL1-TKT heterodimers bind X5P (the substrate for R5P removal in reaction 1) more weakly, but bind S7P (the substrate for R5P production in reaction 3) more tightly than TKT homodimers do. On the other hand, TKTL1-TKT heterodimers and TKT bind the remaining substrates with similar affinities (Fig. 4d).

We then examined TKT activities during different phases of the cell cycle. Releasing double thymidine-synchronized $G_1/S$ and RO3306-synchronized $G_2/M$ HeLa cells showed that reaction 1 TKT activities were relatively low in TKTL1-expressing $G_1$ and S phases and relatively high in TKTL1-weakly expressing $G_2$ and M phases (Figs. 4e, 1b, c). Conversely, reaction 3 TKT activities were positively correlated with TKTL1 levels during cell cycle progression (Figs. 4f, 1b, c). These results further support the asymmetric TKT activities of TKTL1-TKT heterodimers.

### TKTL1 depletes cellular TKT and reprograms R5P metabolism.

R5P metabolism should be determined by the activities of the dominant forms of R5P-associated TKTs, namely the TKT homodimer, TKTL1 monomer, and TKTL1-TKT heterodimer. When TKTL1 was progressively overexpressed to a level comparable to that of $G_1/S$ phase (see Fig. 1b) in HeLa cells, removal of ectopically expressed TKTL1 from the cell lysates by affinity purification dose-dependently decreased their TKT levels (Fig. 5a), suggesting that TKTL1-TKT heterodimer formation depletes TKT homodimers from the cells. Moreover, the removal of TKTL1 by an anti-TKTL1 antibody from TKTL1-overexpressing ccRCC tissues almost depleted TKT levels, whereas the removal of TKTL1 from TKTL1-low expressing, corresponding adjacent normal tissues did not deplete TKT (Fig. 5b). These results confirm that TKTL1 depletes TKT in vitro and in vivo.

To investigate whether these observed fluctuations in the TKTL1/TKT abundance ratio—approximately 3.5 times higher in $G_1/S$ than in $G_2/M$ phases in HeLa cells (Fig. 5c)—are sufficient to regulate R5P levels, we progressively overexpressed TKTL1 in HeLa cells and measured cellular R5P levels. A nearly four-fold TKTL1 overexpression of TKTL1 in HeLa cells caused an approximate 100% increase in cellular R5P levels (Fig. 5d). Moreover, compared with corresponding adjacent non-cancer tissues, relative TKTL1 levels in ccRCC tissues were nearly 3 times higher (Fig. 5e), consistent with the almost doubled R5P levels (see Fig. 1i). These results confirm that cell cycle-dependent fluctuations in TKTL1 levels (See Fig. 1b, c) are sufficient for mediating cellular R5P levels.

We further studied the consequences of TKTL1-TKT heterodimer-mediated reprogramming of R5P metabolism. We examined the composition of R5P in cells under different TKTL1 levels by tracing the formation of R5P from $[1,2-^{13}C]$ glucose, which results in both single $^{13}C$-labeled (M1) R5P from oxidative PPP and double $^{13}C$-labeled (M2) R5P from non-oxidative PPP (Fig. 5f). TKTL1 overexpression (Fig. 5g)—but not overexpression of the truncated TKTL1$^{118-448}$ (Supplementary Fig. 6a) that does not bind to TKT (see Fig. 3g)—increased total R5P levels. This increase was due to an increased production of M2 R5P, not M1 R5P (Fig. 5g). Moreover, S7P, the R5P precursor from non-oxidative PPP, decreased due to TKTL1 overexpression (Fig. 5g). However, TKTL1$^{118-448}$ overexpression in HeLa cells did not induce similar changes (Supplementary Fig. 6). These results support the model of TKTL1-TKT heterodimer formation reprogramming R5P metabolism.

Compared with TKTL1 wild-type cells, TKTL1 D-box mutant cells with elevated TKTL1 protein levels led to increased non-oxidative PPP-derived M2 R5P and total R5P concentrations (Fig. 5h). Notably, TKTL1 overexpression in HeLa cells that were TKT-silenced to prevent TKTL1-TKT heterodimer formation failed to alter R5P levels and the M1 and M2 R5P ratios (Fig. 5g). Moreover, HeLa cells entering the high TKTL1-expressing $G_1$ phase had increased levels of total R5P, predominantly due to an increase in M2 R5P (Fig. 5g). Meanwhile, cells entering S phase had decreased total R5P levels, mainly due to a drop in M2 R5P levels (Fig. 5i); this is consistent with a previous study where the majority of R5P incorporated into de novo and salvage purine synthesis in S phase was reported to originate from non-oxidative PPP[23]. However, R5P and M2 R5P levels in TKT-knockdown HeLa cells were not dramatically altered compared with those of HeLa cells during the cell cycle (Fig. 5j). Furthermore, knockout of the key glycolytic enzyme PFKFB3—also regulated by APC/C$^{CDH1}$ and a limiting factor in providing the G3P that links

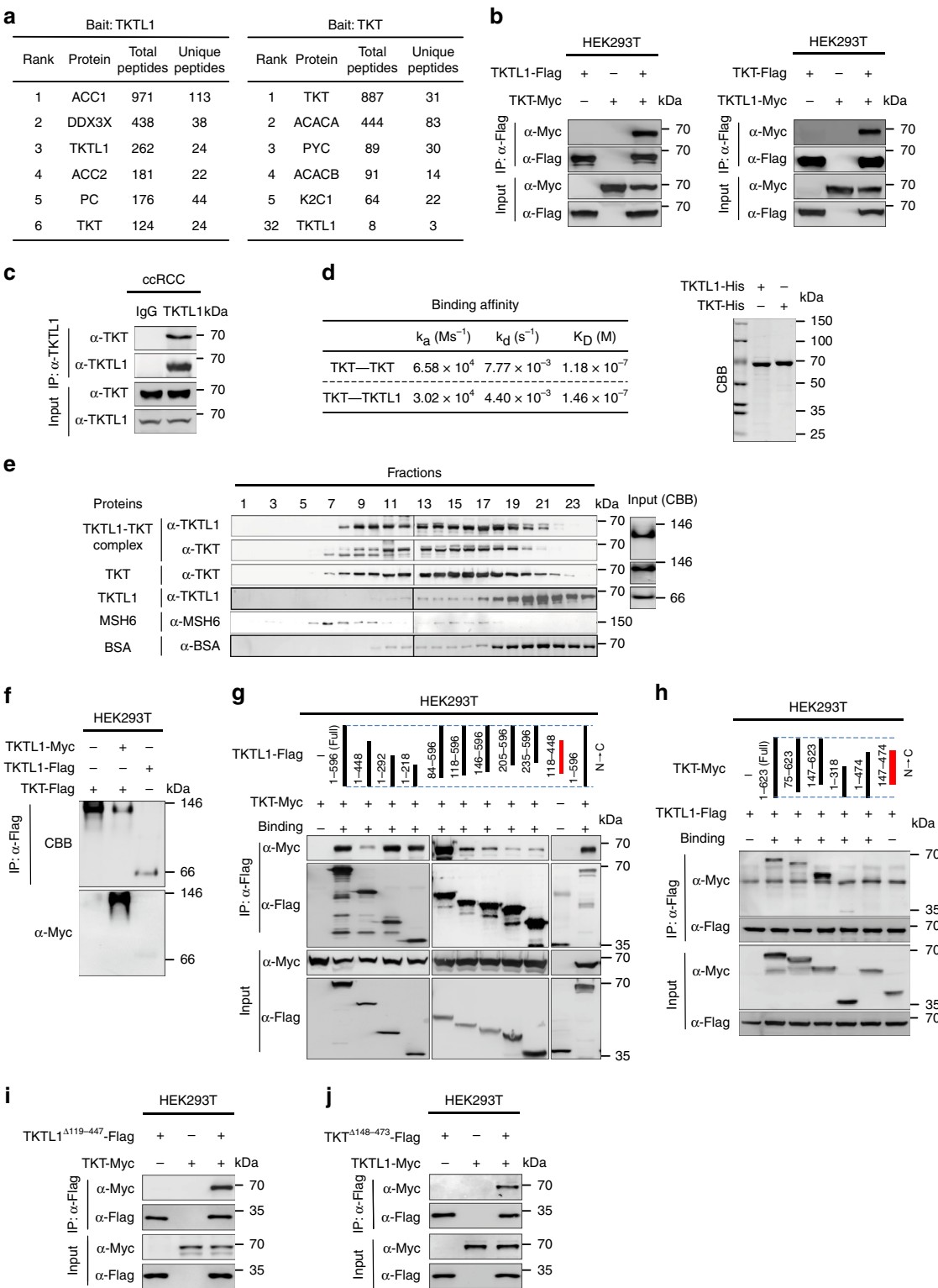

**Fig. 3** TKTL1 depletes TKT by forming stable heterodimers with TKT. **a** Interacting proteins of TKTL1 and TKT. Tandem affinity purification identification of TKTL1- (left) or TKT-interacting (right) proteins. **b** TKTL1 binds to TKT. TKTL1 and TKT associations were examined by reciprocal co-immunoprecipitation as indicated. **c** Affinity purified TKTL1 from ccRCC and co-purification of endogenous TKT. **d** Binding affinity of TKT to TKTL1 and TKT was analyzed by SPR. The right panel shows the purity of both TKTL1 and TKT. CBB, Coomassie brilliant blue. **e** Gel filtration assays for determining heterodimers MSH6 (152 kDa) and BSA (66 kDa) proteins were used as molecular weight markers. The right panel shows the purity of both TKTL1 and TKT. **f** Native gel electrophoresis for detecting formation of the TKT dimer, TKTL1-TKT heterodimer, and TKTL1 monomer. **g**, **h** (**g**) TKT was co-expressed with sequentially truncated TKTL1 species and (**h**) TKTL1 was co-expressed with sequentially truncated TKT species in HEK293T cells; interactions between the overexpressed proteins were analyzed by co-immunoprecipitation. Binding (+) and no binding (−) between species are indicated. **i**, **j** (**i**) TKTL1 without the central portion was co-expressed with TKT and (**j**) TKT without the central portion was co-expressed with TKTL1; interactions between the overexpressed proteins were analyzed by co-immunoprecipitation. Full-length blots are presented in Supplementary Fig. 10

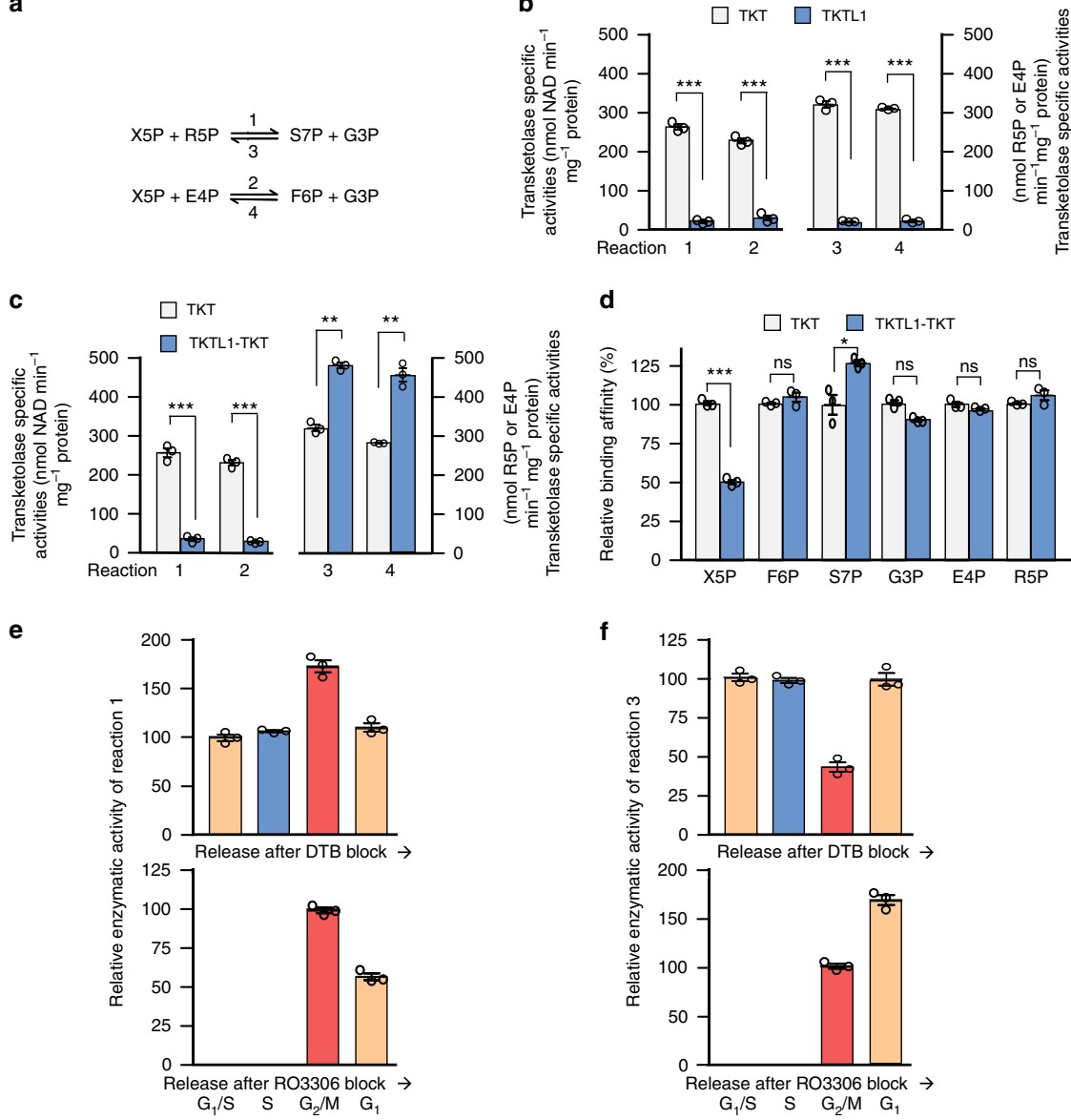

**Fig. 4** The TKTL1-TKT heterodimer possesses asymmetric transketolase activities. **a** Schematic diagram of the four transketolase activities. **b** The absolute specific transketolase activities of recombinant TKTL1 and TKT were determined and then compared. Data are shown as average of 3 independent experiments, presented as means ± SEM, two-tailed Students' t test, ***$p < 0.001$. **c** Specific transketolase activities of the recombinant TKTL1-TKT heterodimer and TKT-TKT homodimer were determined and then compared. Data are presented as means ± SEM of three independent experiments, two-tailed Student's t test, ***$p < 0.001$, **$p < 0.01$, *$p < 0.05$. **d** Substrate binding to the TKTL1/TKT heterodimer and TKT homodimer were analyzed via SPR. Mean values with SEM are reported, Student's t test, ***$p < 0.001$, **$p < 0.01$, *$p < 0.05$, ns not significant vs the corresponding control group. **e, f** Relative (to total transketolase activity) R5P removal activity (reaction 1; **e**) and R5P production activities (reaction 3; **f**) at different phases of the cell cycle were determined. Different cell cycle phases were achieved by releasing double thymidine (upper panel) and RO3306 (lower panel)-synchronized HeLa cells. Data are presented by means ± SEM of three independent experiments

glycolysis with non-oxidative PPP—led to decreased total R5P and M2 R5P production in cells (Fig. 5k) and saturated the influence of CDH1 on non-oxidative PPP-derived M2 R5P synthesis (Fig. 5l). These results further support that TKTL1-TKT heterodimer formation is essential for R5P metabolism reprogramming, which accumulates R5P from non-oxidative PPP.

**APC/C^CDH1 regulates nucleotide levels in cell cycle.** We next explored whether TKTL1 regulates the levels of select R5P-containing precursors for DNA synthesis, namely PRPP, IMP,

AMP, and GMP. TKTL1 overexpression in HeLa cells increased total levels and percentage of M2-form R5P-containing molecules but did not alter the percentage of M1-form R5P-containing molecules (Fig. 6a), indicating that increased M2 R5P from the non-oxidative PPP (see Fig. 5f) drove the synthesis of R5P-containing molecules. Notably, CDH1 knockdown phenocopied the effects of TKTL1 overexpression in regulating R5P-containing molecule levels (Fig. 6b), consistent with the observation that CDH1 knockdown increased TKTL1 levels (Fig. 2j). Moreover, CDH1 knockdown in HeLa cells elevated R5P levels but decreased the variations of R5P among $G_1/S$, $G_2/M$, and the following $G_1$ phase (Fig. 6c). R5P levels in S phase were not

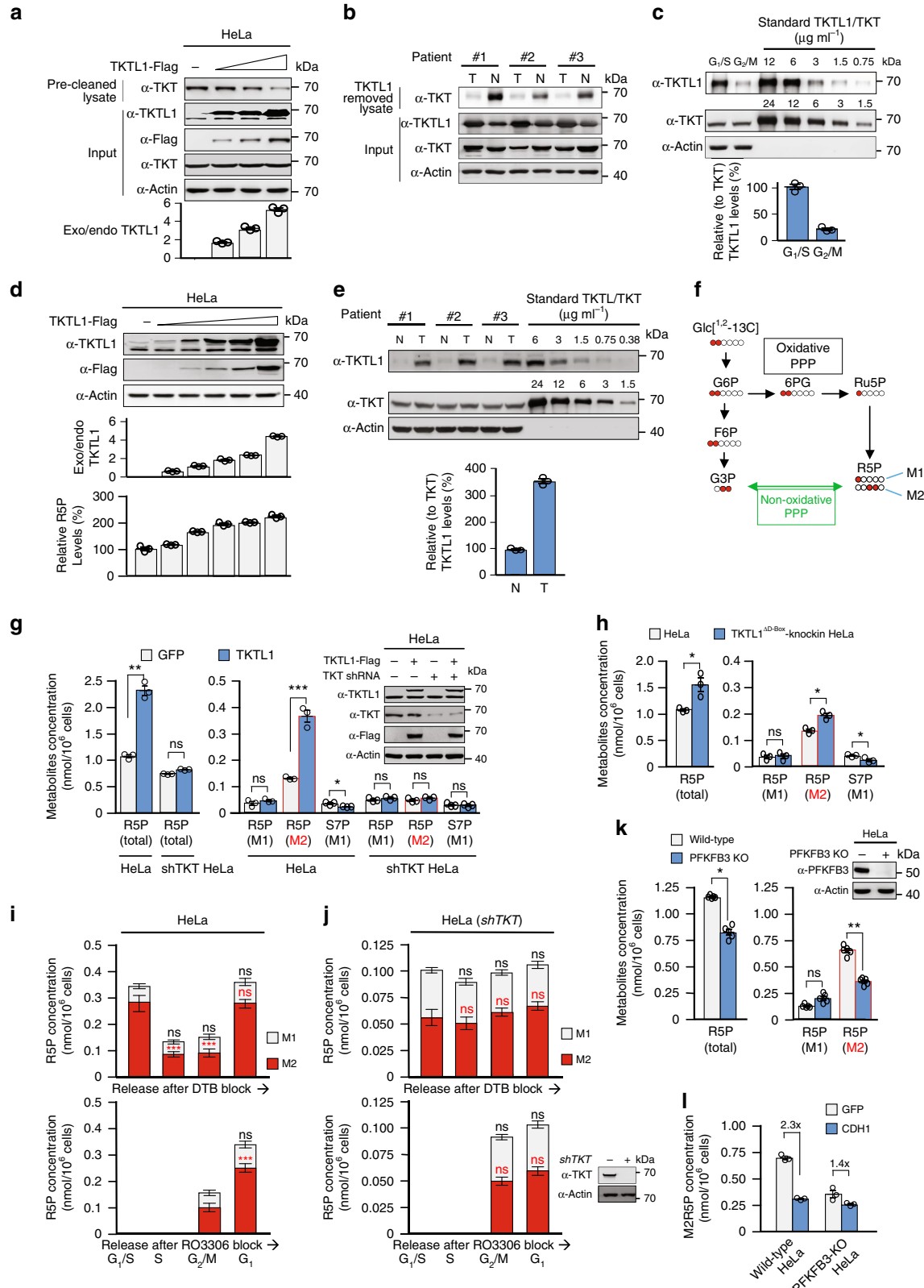

affected by CDH1 knockdown due to the lack of CDH1 expression during S phase (Fig. 6c). Forced expression of CDH1 decreased R5P levels, but R5P levels varied greatly between $G_1/S$, S, and subsequent $G_1$ phases (Fig. 6d). Together with the finding that both CDH1 overexpression and knockdown did not alter R5P levels in TKTL1$^{\Delta D\text{-box}}$-knockin HeLa cells (Supplementary

Fig. 6b, 6c), our results indicate that R5P-containing precursors of DNA synthesis are under regulation by APC/C$^{CDH1}$.

TKTL1 knockdown not only lowered the levels of R5P-containing molecules, but also abolished the effects of CDH1 overexpression in HEK293T cells (Fig. 6e) and CDH1 knockdown in HeLa cells (Fig. 6f), thereby altering the levels of

**Fig. 5** TKTL1 reprograms R5P metabolism. **a** Progressive overexpression of TKTL1 in HEK293T cells and detection of TKT-TKT homodimers in cell lysates after removal of ectopically expressed TKTL1. The ratio of exogenous TKTL1 to endogenous TKTL1 is shown in the lower panel. **b** Affinity removal of TKTL1 from lysates of ccRCC tissues and their corresponding adjacent non-cancer tissues were achieved by TKTL1 antibodies; the remaining TKT levels in cleared lysates were determined. **c** Quantitation of relative TKTL1 levels at different cell cycle phases. The absolute levels of TKTL1 and TKT at $G_1$/S and $G_2$/M phases were calibrated to corresponding recombinant proteins. Data are presented by means ± SEM of three independent experiments. **d** TKTL1 was progressively overexpressed in HeLa cells (upper panel) and then relative (to endogenous) TKTL1 overexpression levels (middle panel) and corresponding cellular R5P levels were determined (lower panel). Data are shown by means ± SEM of three independent experiments. **e** Quantitation of relative TKTL1 levels in ccRCC tissues and their corresponding adjacent non-cancer tissues. The absolute levels of TKTL1 and TKT were calibrated to corresponding recombinant proteins. Data are presented by means ± SEM of three independent experiments. **f** Schematic diagram of the formation of the M1-form R5P from oxidative PPP (black) and M2-form R5P from non-oxidative PPP (green) when [1,2-$^{13}$C] glucose was used as tracer. **g** Total R5P, M1 and M2 R5P, and S7P concentrations were determined in TKTL- or GFP-overexpressing HeLa cells and TKT-knockdown HeLa cells ($n = 3$ biologically independent experiments). The right panel shows the efficiency of TKTL1 overexpression and TKT knockdown. Mean values with SEM are reported, Student's $t$ test, ***$p < 0.001$, **$p < 0.01$, *$p < 0.05$, ns not significant vs the corresponding control group. **h**. Total R5P, M1 and M2 R5P, and S7P levels were compared between HeLa and TKTL1$^{\Delta D\text{-box}}$-knockin HeLa cells. Data are presented as mean ± SEM of three independent experiments, Student's $t$ test, *$p < 0.05$, ns not significant vs the corresponding control group. **i, j** The total (full-length bar), M1 (white-colored bar), and M2 (red colored bar) concentrations of R5P during cell cycle progression in (**i**) HeLa and (**j**) TKT-knockdown HeLa cells were determined. Five biological independent experiments are used to present the data. Student's $t$ test, ns not significant vs the corresponding control group. **k** Total, M1, and M2 R5P concentrations in HeLa and PFKFB3-knockout HeLa cells. Data are shown by means ± SEM of five independent experiments, Student's $t$ test, ***$p < 0.001$, **$p < 0.01$, *$p < 0.05$, ns not significant vs the corresponding control group. **l** M2 R5P concentrations in CDH1- or GFP-overexpressing HeLa cells and PFKFB3-knockout HeLa cells. Data are shown by means ± SEM of three independent experiments, Student's $t$ test, ***$p < 0.001$, **$p < 0.01$, *$p < 0.05$, ns not significant vs the corresponding control group. Full-length blots are presented in Supplementary Fig. 10

R5P-containing molecules. Moreover, the regulating effects of CDH1 knockdown (Fig. 6g) and overexpression (Fig. 6h) were also diminished by TKTL1 knockdown. These findings confirm that APC/C$^{CDH1}$ regulates the levels of R5P-containing molecules via TKTL1. Notably, we found that TKTL1 overexpression increased PRPS transcription in an R5P-saturable manner (Fig. 6i), supporting the model of TKTL1 positive regulation of R5P levels and indicating that the increase in R5P-related metabolite levels induced by TKTL1 overexpression is—at least partially—associated with PRPS activation.

**APC/C$^{CDH1}$ regulates DNA synthesis by controlling R5P**. We further explored whether APC/C$^{CDH1}$ and TKTL1 are involved in DNA synthesis and cell cycle regulation. By utilizing 5-ethynyl-2′-deoxyuridine (EDU) staining to monitor DNA synthesis[49], we found that CDH1 knockdown increased while CDH1 overexpression decreased DNA synthesis in HEK293T cells. However, CDH1 exerted DNA synthesis regulating effects only when TKTL1 was present in the cells; depleting TKTL1 abrogated CDH1 ability to regulate DNA synthesis (Fig. 7a and Supplementary Fig. 7a). These results are consistent with the finding that APC/C$^{CDH1}$ regulated DNA synthesis by controlling TKTL1. Moreover, TKTL1 knockdown and TKTL1 overexpression decreased and increased DNA synthesis, respectively, in HEK293T cells (Fig. 7b); consistent with the finding that CDH1 negatively regulated TKTL1 (see Fig. 2). Furthermore, supplementation with 10 mM R5P promoted DNA synthesis and saturated the DNA synthesis-promoting effects of CDH1 knockdown or TKTL1 overexpression (Fig. 7a, b), confirming that both CDH1 knockdown and TKTL1 overexpression promoted DNA synthesis by increasing R5P levels. In addition, PFKFB3 knockout decreased DNA synthesis and weakened the influence of CDH1 on DNA synthesis (Supplementary Fig. 7b).

The S phase content (the percentage of S-phase cells), a readout for proliferation activity and cell cycle progression[50], was increased by CDH1 knockdown in HEK293T cells in a R5P saturable manner (Fig. 7c), whereas CDH1 overexpression in HEK293T cells decreased the percentage of S-phase cells in a R5P rescuable manner (Fig. 7d). Additionally, TKTL1 overexpression and knockdown in HEK293T cells increased and decreased the percentage of cells in S phase, respectively (Fig. 7e, f), and both effects were abolished by R5P supplementation (Fig. 7e, f).

Furthermore, TKTL1 knockdown in HEK293T cells abrogated the effects of CDH1 knockdown and CDH1 overexpression in decreasing and increasing S phase content, respectively (Fig. 7c, d), whereas CDH1 knockdown in HEK293T cells failed to affect the inhibition and promotion of S phase content by TKTL1 overexpression and knockdown, respectively (Fig. 7e, f). These results confirm that CDH1 regulates the cell cycle through TKTL1. Notably, TKT knockdown, in which R5P levels were unresponsive to cell cycle progression (Fig. 5j), rendered the S phase content of HeLa cells unresponsive to both TKTL1 overexpression and CDH1 knockdown (Fig. 7g). This suggests that TKTL1-TKT heterodimer formation is required for both CDH1 and TKTL1 to regulate cell cycle progression.

To further confirm the cell cycle-regulating effects of CDH1 and TKTL1, we tested their effects on cell growth. HeLa, HEK293T, and breast cancer MCF7 cell growth was enhanced by CDH1 knockdown and TKTL1 overexpression, and exhibited inhibited growth after CDH1 overexpression and TKTL1 knockdown (Fig. 7h and Supplementary Fig. 8). However, the CDH1 and TKTL1 effects on cell growth were abolished by R5P supplementation, which promoted cell growth of all tested cell lines (Fig. 7h, i, and Supplementary Fig. 8). These results collectively support the existence of an APC/C$^{CDH1}$-TKTL1 regulating pathway that integrates R5P as well as nucleotide and DNA synthesis into cell cycle regulation (Fig. 7j).

## Discussion
DNA duplication in S phase is the last major anabolic process before cell division. Due to the extensive biomass- and energy-consuming nature of DNA duplication, cells must accumulate nucleotides for DNA duplication. In the present study, we revealed that cells proactively accumulate nucleotides and facilitate DNA duplication by upregulating R5P in late $G_1$ and S phase prior to, or during, DNA duplication. Notably, regulation of R5P and nucleotide levels was accomplished by APC/C$^{CDH1}$, a classic cell cycle-regulating complex that is known to establish stable M and $G_1$ phases during cell cycle progression[2]. This establishment of stability causes direct control of R5P and nucleotide levels by cell cycle-regulating factors, making it possible for cell cycle-determining machinery to orchestrate R5P and nucleotide levels —and thus the speed of DNA replication—for cell cycle progression. In late $G_1$ and early S phase when DNA synthesis

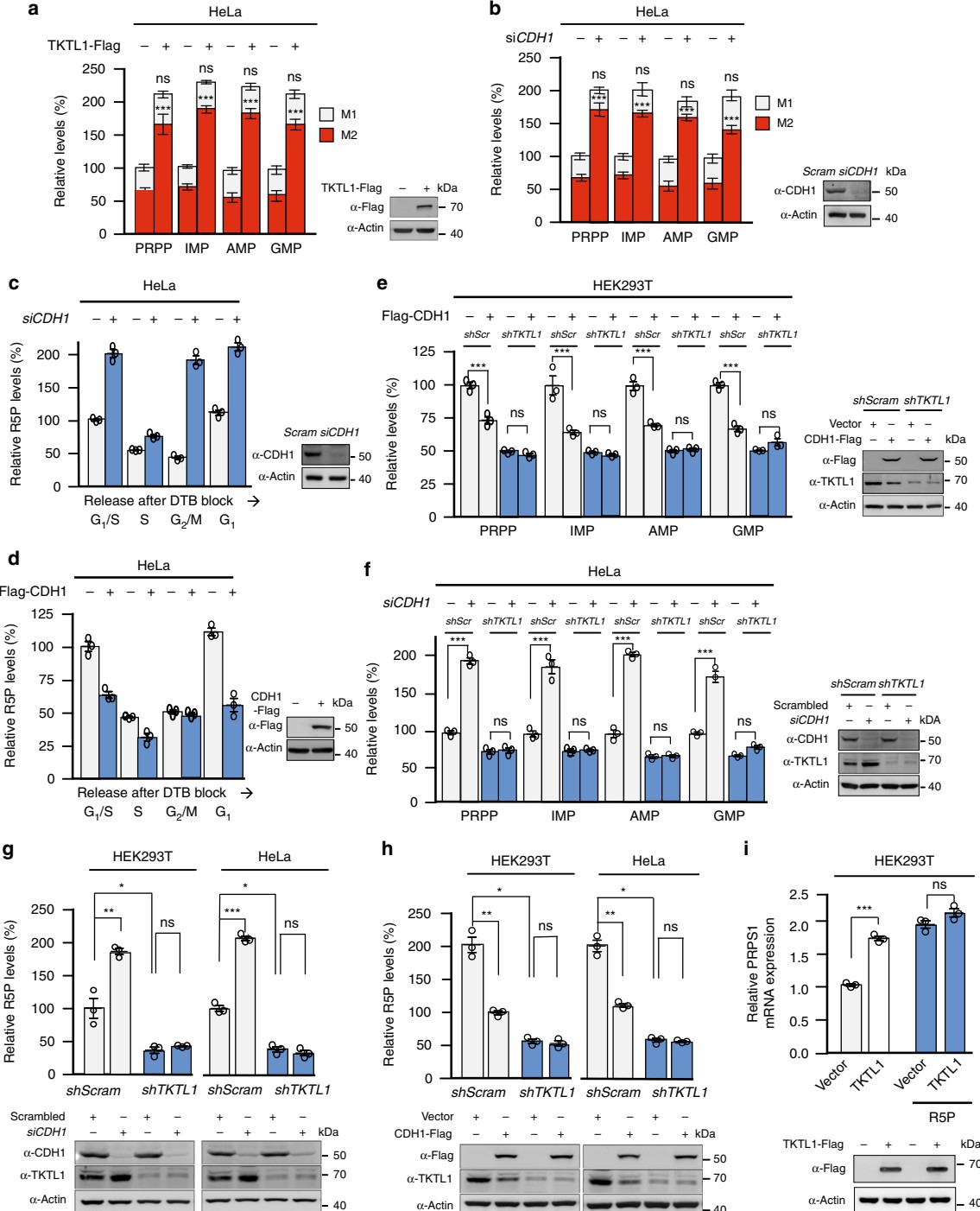

**Fig. 6** APC/C$^{CDH1}$ regulates R5P and nucleotide levels by controlling TKTL1 levels. **a**, **b** Relative levels of total, M1, and M2 R5P-containing metabolites were compared between (**a**) HEK293T and TKTL1-overexpressing HEK293T cells and between (**b**) HEK293T and CDH1-knockdown (siRNA) HEK293T cells. Total levels of R5P-containing metabolites in untreated cells were set as 100%. The significance between level changes in M1 and M2 species was analyzed. Data are presented as mean ± SEM, Student's $t$ test, ns not significant vs the corresponding control group. **c**, **d** R5P levels in HeLa cells were compared with that of (**c**) CDH1-knockout HeLa cells and (**d**) CDH1-overexpressing HeLa cells. Cell cycle phases were achieved by double thymidine blocking followed by release; R5P levels in the initial G$_1$ phase were arbitrarily set as 100%. Data are shown by means ± SEM of three independent experiments. **e**, **f** The effects of CDH1 overexpression and CDH1 knockdown on levels of R5P-containing metabolites were determined in (**e**) HEK293T and TKTL1-knockdown HEK293T cells as well as in (**f**) HeLa and TKTL1-knockdown HeLa cells. Data are presented by means ± SEM of three independent experiments, Student's $t$ test, \*\*\*$p < 0.001$, \*\*$p < 0.01$, \*$p < 0.05$, ns not significant vs the corresponding control group. **g**, **h** The effects of (**g**) CDH1 overexpression and (**h**) CDH1 knockdown on the levels of R5P-containing metabolites were compared in HEK293T and HeLa cells (empty bars) as well as in TKTL1-knockdown HEK293T and HeLa cells (blue bars), respectively. Nucleotide levels in untreated HEK293T and HeLa cells were arbitrarily set as 100%. Data are presented by means ± SEM of three independent experiments, Student's $t$ test, \*\*\*$p < 0.001$, \*\*$p < 0.01$, \*$p < 0.05$, ns not significant. **i** PRPS1 mRNA levels were measured in HEK293T cells, TKTL1-overexpressing HEK293T cells, and TKTL1-overexpressing HEK293T cells cultured in 10 mM R5P-supplemented medium. Data are presented as mean ± SEM, Student's $t$ test, \*\*\*$p < 0.001$, ns not significant vs the corresponding control group. Full-length blots are presented in Supplementary Fig. 10

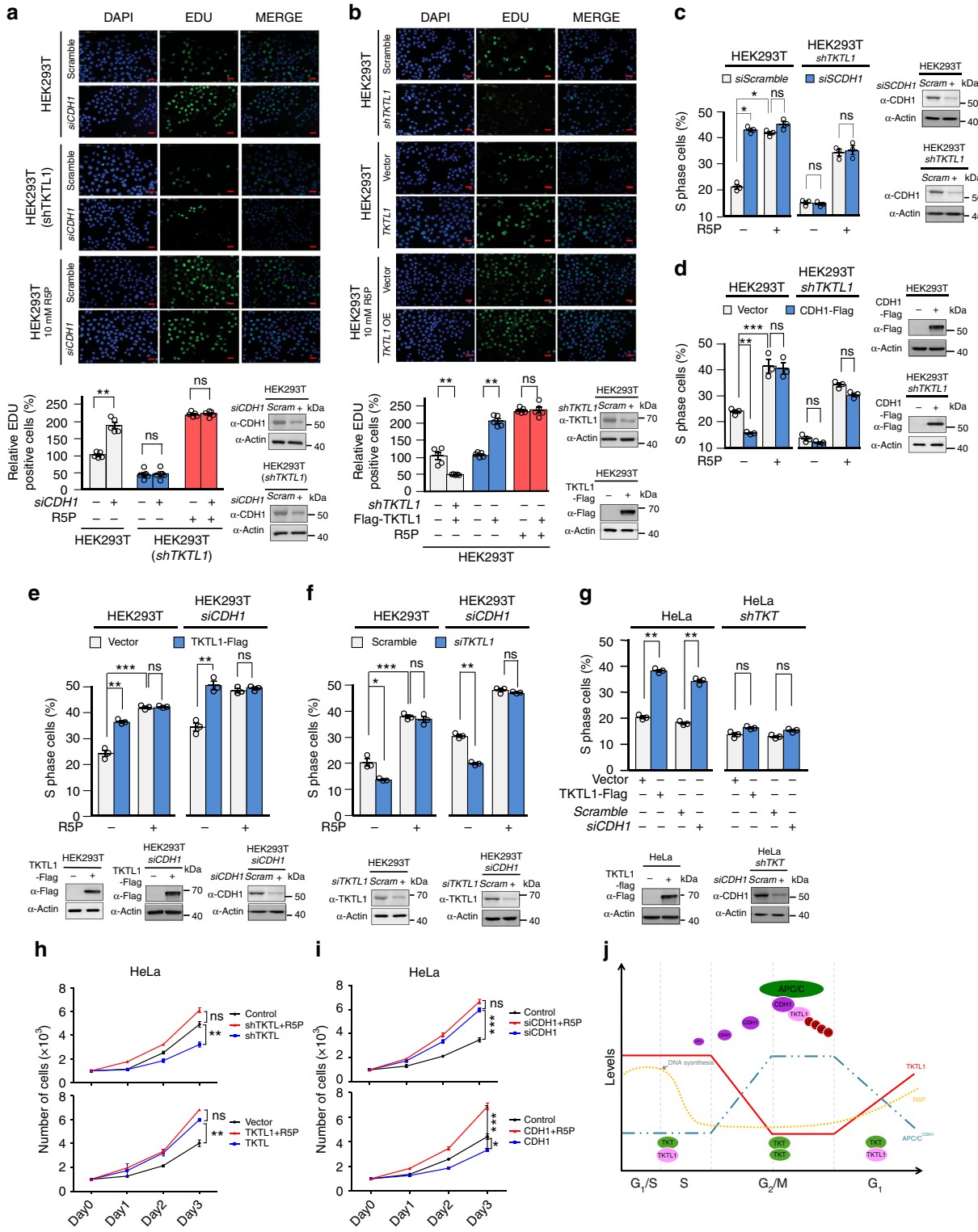

begins, the absence of CDH1 results in increased levels of TKTL1; this reprograms R5P metabolism by forming TKTL1-TKT heterodimers, which produce R5P from the non-oxidative branch of PPP and inhibits R5P removal. This heterodimerization maintains R5P at elevated levels, facilitating synthesis of nucleotides and DNA. During the G$_2$/M phases after DNA duplication, the increase in CDH1 (i.e., APC/C$^{CDH1}$) results in the degradation of TKTL1, allowing TKT to remove R5P, thereby maintaining R5P at low levels to prevent nucleotide synthesis. This pathway

provides a mechanism of facilitating DNA synthesis when required and inhibiting DNA synthesis when not necessary. Our findings are consistent with and explain previous findings where CDH1 knockdown and overexpression elevated and decreased the percentage of S phase content in MCF7 cells, respectively[51], and where TKTL1 overexpression increased S phase content of melanoma LM-MEL-59 cells[52]. Considering that APC/C$^{CDH1}$ also controls the protein stability of PFKFB3, a rate-limiting enzyme of glycolysis[11], we validate PFKFB3 is a limiting factor to

**Fig. 7** APC/C$^{CDH1}$ regulates DNA synthesis and cell cycle by controlling TKTL1 levels. **a** Representative EDU staining results are shown for HEK293T cells, CDH1-knockdown HEK293T cells, CDH1-overexpressing HEK293T cells, and CDH1-knockdown HEK293T cells cultured in 10 mM R5P-supplemented media. Quantitation of EDU staining from five independent assays is shown at the bottom. Data are presented as mean ± SEM, Student's *t* test, ***$p < 0.001$, **$p < 0.01$, *$p < 0.05$, ns not significant vs the corresponding control group. Scale bar = 100 μm. **b** Representative EDU staining results are shown for HEK293T cells, TKTL1-knockdown HEK293T cells, TKTL1-overexpressing HEK293T cells, and TKTL1-overexpressing HEK293T cells cultured in 10 mM R5P-supplemented media. Quantitation of EDU staining from five independent assays is shown at the bottom. Data are shown as mean ± SEM of three independent experiments, Student's *t* test, ***$p < 0.001$, **$p < 0.01$, *$p < 0.05$, ns not significant vs the corresponding control group. Scale bar = 100 μm. **c, d** Effects of (**c**) CDH1 knockdown and (**d**) CDH1 overexpression on S phase content of HEK293T and TKTL1-knockdown HEK293T cells were determined when cells were maintained in either DMEM or DMEM supplemented with 10 mM R5P. Data are presented as mean ± SEM of three independent experiments, Student's *t* test, ***$p < 0.001$, **$p < 0.01$, *$p < 0.05$, ns not significant vs the corresponding control group. **e, f** Effects of (**e**) TKTL1 knockdown and (**f**) overexpression on S phase content of HEK293T and CDH1-knockdown HEK293T cells were determined when cells were maintained in either DMEM or DMEM supplemented with 10 mM R5P. Data are shown as mean ± SEM of three independent experiments, Student's *t* test, ***$p < 0.001$, **$p < 0.01$, *$p < 0.05$, ns not significant vs the corresponding control group. **g** Effects of TKTL1 overexpression and CDH1 knockdown on S phase content were determined in HeLa cells and TKT-knockdown HeLa cells. Data are presented as mean ± SEM of three independent experiments, Student's *t* test, ***$p < 0.001$, **$p < 0.01$, *$p < 0.05$, ns not significant vs the corresponding control group. **h, i** HeLa cell growth was determined under (**h**) TKTL1 overexpression and knockdown and (**i**) CDH1 overexpression and knockdown as well as (**h** and **i**) after culturing in media supplemented with 10 mM R5P. Data are shown as mean ± SEM of three independent experiments, Student's *t* test, ***$p < 0.001$, **$p < 0.01$, *$p < 0.05$, ns not significant vs the corresponding control group. **j** Schematic diagram of the regulation of R5P metabolism and DNA synthesis. TKTL1/TKT heterodimer accumulates R5P and facilitates DNA synthesis at G$_1$ and early S phases, while APC/C$^{CDH1}$ degrades TKTL1 at G$_2$/M phases and allows TKT homodimers to decrease R5P levels and inhibit DNA synthesis. Full-length blots are presented in the Supplementary Fig. 10

provide the necessary G3P that links glycolysis with R5P through non-oxidative PPP (see Fig. 5k, l). These results indicate a direct control of PPP and glycolysis by APC/C$^{CDH1}$ and suggest a possibility that activation of APC/C$^{CDH1}$ may help inhibiting cancer cell progression by prohibiting R5P generation through the inhibition of PPP and early steps of glycolysis.

Proactive regulation of R5P by the APC/C$^{CDH1}$-TKTL1 pathway differs from the current Pulled-In model. Our study also identified a biological function for TKTL1, which was previously undefined[53]. The formation of TKTL1-TKT heterodimers allows cells to reprogram R5P metabolism; when R5P is needed, increased levels of TKTL1 deplete cellular TKT homodimers and enable the TKTL1-TKT heterodimer—with decreased R5P removal capability and increased R5P production ability from non-oxidative PPP—to accumulate R5P. This mechanism provides a molecular basis for a previously puzzling observation where R5P used for nucleotide and DNA synthesis during S phase was found to mainly originate from non-oxidative PPP[23]. Moreover, the reprogramming of R5P metabolism to obtain R5P from non-oxidative PPP by TKTL1 may be advantageous for cells, because the oxidative branch of PPP is the most direct route for NADPH production from glucose[54] and must be maintained at relatively stable levels during the cell cycle. If the oxidative branch of PPP is employed to produce R5P prior to and during S phase, an increase in oxidative PPP activity will cause an increase in NADPH production, which may activate NADPH oxidases and accumulate reactive oxygen species[55,56], exerting deleterious effects on DNA replication.

Cancer cells often overexpress TKTL1, which maintains high R5P levels and facilitates nucleotide synthesis and DNA duplication. In addition, overexpressed TKTL1 leads to acetyl-CoA accumulation which is an energy rich building block for lipid and amino acid synthesis[28]. This overexpression is consistent with the observation that elevated TKTL1 is required for rapid cell growth[57] and that TKTL1 downregulation can inhibit tumor cell growth[44,45]. As TKTL1-TKT heterodimer formation is required for accumulating R5P and accelerated cell proliferation, methods of interfering with TKTL1-TKT heterodimer formation to decrease R5P accumulation and inhibit DNA synthesis should be further explored to contain diseases such as cancer.

## Methods

**Clinical samples.** For western blotting, immunostaining, and metabolites quantification, 30 pairs of tumors and adjacent normal tissues from patients with ccRCC

were collected during surgery between December 2015 and July 2016 at the Affiliated Cancer Hospital of Fudan University. All enrolled patients had been diagnosed with ccRCC for the first time and had not received prior chemotherapy or radiotherapy. The study protocol was reviewed and approved by the ethics committee of the Affiliated Cancer Hospital of Fudan University. Written informed consent forms were obtained from the patients prior to the beginning of the study.

**Plasmids.** Polymerase chain reaction (PCR)-amplified human TKT was cloned into pRK7-FLAG vector between Sal*I* and EcoR*I* and into pcDNA3.1(b+)-MYC between Xho*I* and EcoR*I*. TKTL1 was cloned into pRK7-Flag vector between Hind*III* and EcoR*I* and into pcDNA3.1(b)-Myc between EcoR*I* and Hind*III*. CDH1 was cloned into pRK7-Flag vector between EcoR*I* and Hind*III* and into pcDNA3.1 (b)-Myc between Nhe*I* and EcoR*I*. CDC20 was cloned into pRK7-Flag between Hind*III* and EcoR*I*. SKP2 and FBXW7 were cloned into pRK7-Flag by One step clone kits. The sequences of the primers used in this study are listed in Supplementary Table 1.

For purifying heterodimer proteins of TKT and TKTL1, full-length TKT and TKTL1 were cloned in pET Duet3 vectors obtained from YANHUI XU Laboratory. TKT was inserted between EcoR*I* and Hind*III* after the His-tag, while TKTL1 was cloned into the vector between Nde*I* and Xho*I* with a FLAG-tag.

**Antibodies.** The antibody against for TKTL1 (#NBP1-31674, dilution 1:1000) was purchased from Novus Biologicals. The CDC20 (#4823, dilution 1:3000), SKP2 (#4358, dilution 1:1000) antibody was from Cell Signaling Technology. CDH1 (#CC43, dilution 1:500) was obtained from Millipore. The antibody against TKT (#sc-67120, dilution 1:3000) was purchased from Santa Cruz Biotechnology. RPIA (#181235, dilution 1:1000) antibody was from Abcam. Anti-β-actin (A00702, dilution 1:10,000) antibody was purchased from GeneScript. Anti-Flag (#M20008, dilution 1:5000), Anti-Myc (#M20003, dilution 1:5000), and anti-HA (#M20002, dilution 1:5000) antibodies were obtained from Abmart.

**Chemicals.** DAPI (#D8417) was from Sigma-Aldrich. EdU (#A10044) and Azide Alexa Fluor(#A10266) were purchased from Invitrogen.

**Cell culture and treatment.** HEK293T (ATCC Number: CRL-11268), HeLa (ATCC Number: CCL-2) and MCF7 (ATCC Number: HTB-22) were purchased from Shanghai Cell Bank and tested negative for mycoplasma contamination. HeLa cells were authenticated using Short Tandem Repeat (STR) analysis by Shanghai Biowing Applied Biotechnology Company. HeLa and HEK293T cells were cultured in DMEM (HyClone) supplemented with 10% newborn bovine serum (HyClone), 100 units mL$^{-1}$ penicillin, and 100 μg mL$^{-1}$ streptomycin (Invitrogen). MCF7 cells were cultured in DMEM (HyClone) supplemented with 10% fetal bovine serum (HyClone), 100 units mL$^{-1}$ penicillin, and 100 μg mL$^{-1}$ streptomycin (Invitrogen). For ubiquitination assays, the proteasome inhibitor, MG132, was added 4 h before harvesting the cells.

PFKFB3 knockout HeLa cell lines are kindly provided by Dr. Ye Dan, MCB laboratory, Fudan University. The guide sequence targeting the human PFKFB3 gene is 5′- AGC TGA CTC GCT ACC TCA AC-3′.

**Tandem affinity purification.** 293T cells were transfected with pMCB-SBP-Flag-TKT or TKTL1 containing a puromycin resistance marker. The TKT or

TKTL1-positive stable cells were lysed on ice in 0.1% NP40 buffer containing 50 mM Tris-HCl (pH 7.5), 150 mM NaCl, 0.3% NONIDET P-40, 1 µg mL$^{-1}$ aprotinin, 1 µg mL$^{-1}$ leupeptin, 1 µg mL$^{-1}$ pepstatin, and 1 mM PMSF. After removal of insoluble cell debris by high speed centrifugation, cell lysates were incubated with SBP beads (Millipore) for 3 h at 4 °C. The precipitates were washed three times with 0.1% NP40 buffer, two times with ddH$_2$O, and three times with 50 mM NH$_4$HCO$_3$. On-bead tryptic digestion was performed at 37 °C overnight. The peptides in the supernatant were collected by centrifugation and dried in a speed vacuum (Eppendorf). Samples were re-dissolved in NH$_4$HCO$_3$ buffer containing 0.1% formic acid and 5% acetonitrile (ACN) before being subjected to mass spectrometry.

**Cell transfections, immunoprecipitation, and immunoblotting.** Plasmid transfections were carried out by the Polyethylenimine (PEI), Lipofectamine 2000 (Invitrogen), or calcium phosphate methods. In the PEI transfection method, 500 µL of DMEM (serum-free medium) and the plasmid were placed in an empty EP tube and PEI (three times the concentration of plasmid) was added into the medium with vigorous shaking. The mixture was incubated for 15 min. Meanwhile, the cell culture medium was replaced with 2 mL of fresh 10% NCS medium. After 15 min, the mixture was added to the cells, and the fresh medium was replaced after 12 h. After 36 h, the transfection was completed and the cells were treated or harvested. In the Lipofectamine 2000 transfection method. DMEM (250 µL) was added to two clean EP tubes and 6 µL of Lipofectamine 2000 was added to one of the tubes and mixed for 5 min The plasmid was added in the other tube and then added to the medium containing Lipofectamine 2000, mixed, and allowed to stand for 20 min Meanwhile, the cell culture medium was replaced by serum-free medium as serum interferes with Lipofectamine 2000 transfection efficiency. Six hours after the addition of the mixture to the cells, the medium was replaced with fresh normal medium and cells were collected 36 h later. In the calcium phosphate method, we aspirated the medium and added 9 mL fresh DMEM first, and then placed the cells back into the incubator for at least 1 h. This is important to balance the pH for transfection efficiency. The DNA in ddH$_2$O (up to 450 µL) was mixed with 500 µL of 2 × HBS buffer and 50 µL of CaCl$_2$ was added drop by drop with shaking. The mixture was incubated on ice for 10 min Chloroquine (2000×, 5 µL) was added to the cells and the mixture was added drop by drop into the plates gently. The plates were swirled and placed back into the incubator. After 5–6 h after transfection, the medium was aspirated and the cells were washed twice with PBS and fresh medium was added. The cells were collected 24–48 h later.

For immunoprecipitation, cells were lysed with 0.5% NP-40 buffer containing 50 mM Tris-HCl (pH 7.5), 150 mM NaCl, 0.3% NONIDET P-40, 1 µg mL$^{-1}$ aprotinin, 1 µg mL$^{-1}$ leupeptin, 1 µg mL$^{-1}$ pepstatin, and 1 mM PMSF. Cell lysates were incubated with Flag beads (Sigma) for 3 h at 4 °C. The binding complexes were washed with 0.5% NP-40 buffer and mixed with loading buffer for SDS-PAGE. Full-length blots are presented in the Supplementary Fig. 10.

**Immunohistochemistry.** For immunohistochemical staining, tissue sections were deparaffinized by xylene two times and then hydrated. The sections were blocked with goat serum in TBS for 30 min Each sample was incubated with specific antibodies overnight at 4 °C. The appropriate secondary antibody was then applied and incubated at 37 °C for 1 h. Sections were developed with the DAB kit, and the reaction was stopped with water. The H-score method, which combines the immunoreactivity intensity values and the percentage of stained tumor cells, was used to quantify the positive score of each sample.

**In vivo ubiquitination assay.** HA-tagged ubiquitin and target plasmids were transfected for 36 h. MG132 was added for 4 h before harvesting the cells to inhibit proteasome activity. Cells were next lysed in 1% SDS buffer (Tris-HCl pH 7.5, 0.5 mM EDTA, 1 mM DTT), and boiled for 10 min When the cell pellets became clear, the lysates were diluted 10-fold in Tris-HCl Buffer. Flag Beads (Sigma) was used for immunoprecipitation. Ubiquitination was analyzed by immunoblotting using anti-HA and anti-Flag antibodies.

**RNA interference.** *CDH1* knockdown was carried out using synthetic siRNA oligonucleotides synthesized by Genepharma. A scrambled siRNA was used as a control. For each target gene, we employed two effective target sequences to exclude off-target effects. Transfections were performed by using Lipofectamine 2000 (Invitrogen). The knockdown efficiency was verified by q-RT-PCR or western blotting. Supplementary Table 1 lists the DNA sequences for the siRNA.

To generate cells stably knocked down for TKTL1 and TKT, lentiviruses carrying pMKO empty vector, pMKO-TKTL1, or pMKO-TKT were produced in HEK293T and HeLa cells, using VSVG and GAG as packaging plasmids. The virus supernatant was next collected, to infect target HeLa cells in the presence of 8 µg mL$^{-1}$ polybrene. Puromycin was used to select stable cells for ~5 days. The DNA sequences of the siRNAs are listed in Supplementary Table 1.

**Generation of knockout cells using CRISPR/Cas9 editing.** To generate TKT-knockout HeLa cells, we used the following guide sequence targeting the human TKT, 5′-GGAGCTGATACGTAGGCGGT-3′. To generate TKTL1 D-box knockin

HeLa cells, the guide sequences targeting the TKTL1 were used: (1) 5′-AGGAGG CCAGACCTGACAG-3′; (2) 5′- ATATCTTGCAACACCTGCA-3′.

**TKT activity assay.** To measure TKT enzyme activity in vitro, FLAG-TKT and FLAG-TKTL1 were transfected, immunoprecipitated, and diluted with 250 µg mL$^{-1}$ FLAG peptide.

For forward TKT activity measurement, we coupled measurement of enzyme activity with GAPDH enzymes. Samples were added to a cuvette containing buffer (100 mM K$_2$HPO$_4$ pH 7.6, 5 mM MgCl$_2$, 3 U GAPDH, 2 mM NAD$^+$, and 0.1 mM TPP) and reaction 1 (2 mM ribose 5-phosphate + 1 mM xylulose 5-phosphate) or reaction 2 (2 mM erythrose 4-phosphate + 1 mM xylulose 5-phosphate) substrates. Reactions were initiated by adding TKT or TKTL1 protein samples at 37 °C. The change in absorption of NADH production was measured using an F-4600 Fluorescence Spectrophotometer (Hitachi, Tokyo, Japan) and TKT activity was expressed as nmol NAD$^+$ min$^{-1}$ mg$^{-1}$ protein. Each experiment was repeated three times.

For reverse TKT activity measurement, we measured enzyme activity through detection of either R5P (for reaction 3) or E4P production (for reaction 3) in reversed reactions. Samples were added to a 1.5-mL tube containing buffer (100 mM K$_2$HPO$_4$ pH 7.6, 5 mM MgCl$_2$, and 0.1 mM TPP) and reaction 3 (2 mM sedoheptulose 7-phosphate + 2 mM glyceraldehyde 3-phosphate) or reaction 4 substrates (2 mM erythrose 4-phosphate + 2 mM fructose 6-phosphate). Reactions were initiated by adding TKT or TKTL1 protein samples at 37 °C. R5P and E4P production were measured using LC-MS and TKT activity was expressed as nmol R5P or E4P min$^{-1}$ mg$^{-1}$ protein. Each experiment was repeated three times.

For measuring in vivo TKT activity, cells were sonicated and centrifuged, after which the resulting supernatant was collected for analysis. Buffer containing 100 mM K$_2$HPO$_4$ (pH 7.6), 5 mM MgCl$_2$, 3 U GAPDH, 2 mM NAD$^+$, and 0.1 mM TPP was added to a cuvette. Next, samples were added to the cuvettes and incubated for 60 s. Activity measurements began when 2 mM ribose 5-phosphate and 1 mM xylulose 5-phosphate were added. The change in absorption of NADH production was measured using the F-4600 Fluorescence Spectrophotometer and TKT activity was expressed as ng product min$^{-1}$ mg$^{-1}$ protein. Each experiment was repeated three times.

**Synchronization of HeLa cells.** HeLa cells can be synchronized in G$_1$/S phase via the double thymidine block method. Briefly, HeLa cells of 25–30% density were cultured with 2 mM thymidine medium for 18 h, washed twice with PBS, and placed in fresh 10% NCS DMEM for 9 h. For the second block, cells were cultured with 2 mM thymidine medium for another 17 h. After release from the block, cells could be harvested for western blotting or flow cytometric analysis every 2 h for up to 14 h for whole cell cycle detection. The G$_2$ phase of HeLa cells can be obtained by culturing cells with 9 µM RO3306 (Sigma) for 20 h and 30 min after release, most of the cells are at the G$_2$ phase.

**Flow cytometric analysis.** Approximately 10$^6$ treated cells were suspended in cold 70% ethanol for 3 h, and incubated for 1 h at 37 °C in PBS with DNase-free RNase A (100 mg mL$^{-1}$) and propidium iodide (50 mg mL$^{-1}$). The cells were next analyzed using a fluorescence-activated cell sorter (FACS). The gating strategy of flow cytometry was shown in Supplementary Fig. 9. Data are presented as means ± SD of three independent experiments.

**Cell proliferation assay.** Cells (1 × 10$^3$) were seeded in wells in a E-plate and counted every 10 min over 4 days using RTCA Cardio Instrument following the manufacturer's instructions.

**Recombinant protein purification.** The wild-type TKT and TKTL1 were cloned in vector pSJ3 with 6× HIS tag at the N-terminal. Recombinant heterodimer of TKT and TKTL1 was cloned into the pET-Duet vector. TKTL1 was fused in-frame with 6× HIS tag at the N-terminal, while TKT was inserted in the vector with a Flag tag at the C-terminal. Plasmids were transformed into *E. coli* BL21 (DE3)pLysS strain and protein expression was induced by addition of 0.5 mM IPTG when the cell density reached 0.6 OD$_{600}$ units at 16 °C. Cells were lysed by sonication and nickel columns (GE Healthcare) were used to purify proteins. Heterodimer proteins were isolated by sequential affinity purification using Nickel resin followed by FLAG beads (Sigma-Aldrich).

**Quantitative RT-PCR.** The Superscript III RT kit (Invitrogen) was used with random 3 hexamer primers to produce cDNA from 4 µg total RNA. β−actin was used as the endogenous control for samples. All primers for analysis were synthesized by Generay (Shanghai). Primer sequences are listed in Supplementary Table 1. The analysis was performed by using an Applied Biosystems 7900HT Sequence Detection System, with SYBR green labeling.

**LC-MS/MS measurement.** Approximately 1 × 10$^7$ cells were treated with cold aqueous methanol solution (80% v/v) to quickly stop cell metabolism. Samples were then centrifuged for 15 min at 15,000 × $g$ and 4 °C, after which the

supernatants were collected. The supernatants were then lyophilized and reconstituted in 500 μL methanol/water (10:90 v/v). The separated metabolites were acquired using high-performance liquid chromatography (HPLC) employing an LC-20AB pump (Shimadzu, Kyoto, Japan) and the Luna NH2 column (P/N 00B-4378-B0; 5 μm, 50 × 2.0 mm; Phenomenex, Torrance, CA). The mobile phase comprised eluent A (0.77 g NH$_4$OAc, 1.25 mL NH$_4$OH, 25 mL ACN, and 300 μL acetic acid [HAc] dissolved in 500 mL water) and eluent B (ACN). The elution program was as follows, 0.1 min, 85% B; 3 min, 30% B; 12 min, 2% B; 15 min, 2% B; and 16–28 min, 85% B. The flow rate of the pump was 0.3 mL min$^{-1}$ and the mass spectrometer used was the 4000 QTRAP system (AB Sciex, Framingham, MA) operated in multiple reaction monitoring (MRM) mode. The MS parameters were electrospray voltage, 5 kV; gas 1, 30; gas 2, 30; curtain gas, 25; and temperature, 500. Glyceraldehyde-3-p and dihydroxyacetone phosphate ions were monitored at 169-97 (precursor-product); ribose-5-p and xylulose-5-p ions at 229-97; sedulose-7-phosphate ions at 289-97; erythrose-4-phosphate ions at 199-79; fructose-6-phosphate ions at 258.7; IMP ions at 347-79; AMP ions at 346-79; GMP ions at 362-79; and PRPP ions at 389-291. In order to separate sugar isomers such as R5P, Ru5P, and X5P, a versatile, convenient, and highly selective LC-MS/MS method using tributylamine as a volatile ion pair reagent was employed. Briefly, 10 μL tributylamine was injected into mobile phase flow and R5P, Ru5P and X5P standards were used to indicate the retention time. Each measurement was obtained at least in triplicate.

For 1,2-$^{13}$C$_2$-labeled glucose detection, 5 mM 1,2-$^{13}$C$_2$ glucose and 20 mM non-labeled glucose were used to treat cells for 1 h. The cells were washed twice with PBS after collection to completely remove labeled glucose that was not metabolized by the cells. Ions were monitored via HPLC-MS/MS, as described above; M1 of R5P at 230-97, IMP at 348-79, AMP at 347-79, GMP at 363-79, and PRPP at 390-292. M2 of R5P at 231-97, IMP at 349-79, AMP at 348-79, GMP at 364-79, and PRPP at 391-293. For absolute R5P concentration measurement, 0, 5, 10, 50, 100, 150, and 200 nmol R5P from cell lysis were used to obtain an R5P standard curve. After R5P detection via LC-MS, the absolute concentration of R5P was calculated using the R5P standard curve.

**Gel filtration**. The gel filtration column (Superdex 200; Amersham Biosciences) was washed and equilibrated by cold PBS (4 °C). Proteins were passed over the gel filtration column. The flow speed rate was 0.4 mL min$^{-1}$. Fractions were collected every 0.5 mL per tube and analyzed by western blot. Molecular mass was determined by Gel Filtration Calibration Kit HMW (GE Healthcare).

**EDU staining**. HeLa cells were cultured at an appropriate concentration for growth and 20 μM EDU were added to the cell culture medium for 1 h. Cells were harvested and washed with PBS twice to remove the remaining medium. Paraformaldehyde (4%) was used to fix the cells at room temperature; 0.5%Triton X-100 in PBS was added and incubated for 20 min at room temperature. The cocktail (PBS: 215 μL, 100 mM CuSO$_4$: 10 μL, 2 mM Azide: 0.6 μL, 1 M Sodium Ascorbate: 25 μL) was added for 30 min at room temperature in the dark. DAPI was subsequently added, for nuclear staining. Results were acquired in flow cytometer or cells were observed under a fluorescence microscope.

**Surface plasmon resonance**. The binding kinetics and affinity of TKT with TKTL1 protein or small molecules were analyzed by SPR (Biacore T200, GE Healthcare). The purified soluble 5 mg mL$^{-1}$ TKT protein was covalently immobilized to a CM5 sensor chip via amine group in 10 mM sodium acetate buffer (pH 5.5). To determine the K$_d$ of TKT and TKTL1, TKTL1 protein was diluted to a series of concentrations starting at 65.9 nM. SPR experiments were run at a flow rate of 30 mL min$^{-1}$ in PBS buffer. To detect the binding affinity of small molecules and proteins, small molecules were diluted to 100 μM solutions. The procedure was performed following the manufacturer's instructions.

**Statistical methods**. Statistical analysis was performed using Prism 6.0 software (GraphPad Software, Inc., San Diego, CA, USA.), Excel (Microsoft Corp., Redmond, CA, USA), and R version 2.17. Two-tailed Student's $t$-test was performed for the two-group analysis.

**Reporting summary**. Further information on research design is available in the Nature Research Reporting Summary linked to this article.

## Data availability

All data and genetic material used in this paper are available from the authors on request. The mass spectrometry proteomics data have been deposited to the ProteomeXchange Consortium via the PRIDE partner repository with the dataset identifier PXD013309 [http://proteomecentral.proteomexchange.org/cgi/GetDataset?ID=PXD013309].

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

## Acknowledgements

This work was supported by Grants from the State Key Development Programs of Basic Research of China (Nos., SQ2018YFC100242, 2015CB943300), the National Natural Science Foundation of China (Nos. 31871432, 81771627, 31521003, 31821002, 91753207, 81722021, 81471454, 31671483, 31671453, 31425008), Science and Technology Municipal Commission of Shanghai, China (16JC1405300, 18QA1400300), Shanghai Medical Center of Key Programs for Female Reproductive Diseases (2107ZZ01016), and grant from Key Laboratory of Reproduction Regulation of NPFPC.

## Author contributions

J.Y. Zhao and S.M. Zhao conceived the concept, designed and supervised the experiments; Y.L., C.F.Y., F.J.X., Y.Y.Q., J.T.L., Y.L., Z.L.C., P.C.L. and W.X. performed the experiments; Y.Y.Q. collected the clinic samples. J.Y. Zhao and S.M. Zhao wrote the manuscript. All authors read and discussed the manuscript.

## Additional information

**Competing interests:** The authors declare no competing interests.

