## [Peer Review File · Nature Communications]

Reviewers' comments:

Reviewer #1 (Remarks to the Author):

This manuscript by Li et al. et al. suggest that R5P is regulated in a cell cycle-specific manner by APC-mediated proteolytic turnover of TKTL1. Overall, the cell cycle fluctuation of TKTL1 and its corresponding impact on R5P levels are convincing. The data are also strong in supporting a role for ubiquitin-mediated proteolysis in generating TKTL1 periodicity. It is also clear that TKT levels impact R5P levels, probably through the TKTL1-TKT heterodimer. However, the authors' proposed model that TKTL1 levels are regulated by the APC/CCDH1 is not fully supported by the data. Many experiments presented show a consistent negative correlation between CDH1 levels and TKTL1 levels, but the evidence of a direct causal relationship is lacking. Since altering CDH1 levels could have numerous indirect effects on TKTL1, it is not possible to tell whether CDH1 directly regulates TKTL1 levels, and more experiments need to be done to thoroughly test this hypothesis. The authors write, "exogenous TKTL1 was found to interact with the SCF adaptors SKP2 and WD-40 domain protein 7 (FBXW7)," but they do not pursue this further since it did not repeat in the clear cell renal cell carcinoma (ccRCC) lines that they use throughout the paper. The potential involvement of the SCF in regulating TKTL1 levels may be worth investigating further given the incompatibility of some data with the hypothesis that the APC targets TKTL1.

All that said, this is an interesting manuscript. While there have previously been reports of the APC and SCF in targeting metabolic enzymes, this analysis is quite extensive and the nature of the branched and convergent metabolic pathways made this study much more difficult than those on RNR, for example. In the end, whether it is the APC or the SCF doesn't change this importance of the work, but we need to know for sure.

Specific problems with the APC data.

1. In figure 2H, overexpression of CDH1 appears to have no effect on TKTL1 levels. Similarly, knockdown of CDH1 in panel I of the same figure does not lead to an increase in TKTL1 levels. This is in contrast with later panels (e.g. figure 2J and 2N) where altering CDH1 levels appears to have a subtle effect on TKTL1 levels, and which the authors use as evidence that CDH1 directly regulates TKTL1. Similarly, the cycling of TKTL1 in panel 1B is not seen in figure 2O. The fact that data shows the desired result in the panels in which the authors wish to make a point, but not in the subsequent panels, is troubling.
 2. The in vitro ubiquitination pattern of TKTL1 in figure 2K is very unusual.
 3. In figure 2L the authors claim that the difference in binding between CDH1 and wild-type versus D-box mutant TKTL1 shows that CDH1 binds TKTL1 through its D-box. However, the binding difference between wild-type and mutant is subtle, and, more importantly, the input level of the D-box mutant is actually less than the wild-type. Therefore, it is not clear that there is a loss of binding in the D-box mutant. Since this is a crucial experiment for their conclusion that CDH1 regulates TKTL1 levels, this is concerning. While the quantification of the data in Figure 2J appears to show a difference, this is not consistent with the gel that is shown.
 4. Figure 2M is either mis-labelled or improperly performed since the combinations of tagged proteins do not make sense. As is, the experiment does not compare ubiquitination of wild-type TKTL1 to the D-box mutant. This is a crucial experiment for their conclusion that CDH1 regulates TKTL1 levels, so the problem with this figure is concerning. Why is there so much ubiquitination in the D box mutant?
 5. If TKTL1 levels are directly regulated by the APC/CCDH1, then many of the R5P measurements in figure 6 and cell growth assays in figure 7 should be done with a TKTL1 D-box mutant to more directly test this regulation. If TKTL1 is a substrate of the APC/CCDH1, then disrupting the binding between the substrate and the APC should result in altered R5P levels. Although the experiments in figure 6 with CDH1 knockdown and overexpression show a consistent negative correlation between CDH1 levels and TKTL1 activity (measured by R5P and other metabolites), these only indirectly show an association between TKTL1 activity and the cell cycle and do not prove that CDH1 directly regulates TKTL1. As shown by their own data and that of others, altering CDH1 levels can alter the ratio of cells in cell cycle stages, which they show alters TKTL1 levels. I know of many cases where overexpression of CDH1 will strongly alter turnover that is SCF-mediated by altering cyclin levels. Specifically, an experiment like that in 1B, 1C, 5G, and 6C should be completed on a CRISPR generated endogenous D box mutant cell line.
- ADDITIONAL, LESS CRITICAL POINTS ON THE FIGURES.
6. In figure 1E, the flag-tagged proteins are not specifically identified, and neither the text nor the figure legend explicitly explain what the anti-Flag is looking at.
 7. The binding domain identification experiments in figures 3G and 3H are not very informative since the authors need to delete significant domains of TKT and TKTL1 to see loss of binding

between the two proteins. These deletions are large enough that the remaining fragment may no longer fold, so the loss of binding does not support the idea that the absent domains are directly responsible for binding. If they want to support the hypothesis that these termini are sufficient for binding, they need to do that and remove the central portion of the molecule and show that they still bind.

8. For the transketolase assays in figure 4 and the M1/M2 assays in figure 5, it would be informative to include absolute enzymatic activities and R5P levels in addition to the relative values presented in the figure.

9. In figure 5, the authors add increasing amounts of TKTL1 to cells and lysates to show that this depletes TKT, suggesting that the TKTL1-TKT heterodimer is present in vivo. However, the authors do not comment on whether the exogenous TKTL1 levels—particularly in figures 5A and 5D—are comparable to endogenous levels that would actually occur at some stage of the cell cycle.

- Minor point: some of the graph labels in figure 7H are cut off
- There are a number of grammatical errors and confusing sentences in the text.
- The description of the CYHX assay “Inhibition of protein translation by cycloheximide (CHX) failed to prevent TKTL1 levels from decreasing over time (Figure 2A)” while true, is very odd.
- The species is not capitalized. E coli not E. Coli (page 11).

--

Reviewer #2 (Remarks to the Author):

This is an interesting and novel study in which Li and colleagues report the control of ribose-5-phosphate (R5P) levels by the regulation of the transketolase-transketolase-1-like protein (TKT-TKTL1) heterodimer formation. When TKTL1 is high and forms this complex, the formation of the TKT-TKT homodimer is prevented hence avoiding R5P removal from the forward non-oxidative branch of the pentose-phosphate pathway (PPP). In contrast, when the TKT-TKTL1 heterodimer is formed it facilitates the supply of R5P from glycolytically-derived carbon atoms through the reverse non-oxidative branch of the PPP. They also report that this effect is regulated by controlling TKTL1 stability by APC/C-CDH1 through a TKTL1 D-box. In this way, APC/C-CDH1 synchronizes cell cycle progression with R5P-derived DNA synthesis.

The work is elegantly performed and the results are convincing. However, the authors have not put their data in the context of the very well-known role of APC/C-CDH1 on the control of glycolysis (The bioenergetic and antioxidant status of neurons is controlled by continuous degradation of a key glycolytic enzyme by APC/C-Cdh1.

Herrero-Mendez A, Almeida A, Fernández E, Maestre C, Moncada S, Bolaños JP. *Nat Cell Biol.* 2009 Jun;11(6):747-52. doi: 10.1038/ncb1881. Epub 2009 May 17. PMID: 19448625). By the way, the authors should acknowledge this NCB 2009 paper in the last sentence of the first paragraph of the introduction (ref. 10) because it was the first to show the regulation of PFKFB3 (hence glycolysis and PPP activity) by APC/C-CDH1, instead the PNAS-2010 that was erroneously cited.

In fact, PFKFB3 (which is stabilized when CDH1 is low; see Herrero-Mendez et al., 2009) is a limiting factor to provide the necessary G3P that links glycolysis with R5P through the TKTL1-TKT heterodimer. It is therefore surprising that this tight link between the role of APC/C-CDH1 in controlling glycolysis has not been integrated in the message of this study. By doing so, this work could greatly enhance the interest to a wider readership and might support the possibility that inhibition of early steps of glycolysis may help stopping cancer cell progression by prohibiting carbon atoms supply to provide R5P.

Accordingly, it would be nice if the authors could provide any experimental evidence that PFKFB3, which is controlled by APC/C-CDH1 activity (Herrero-Mendez et al., 2009), is actually playing a key role in providing substrate availability for the reverse non-oxidative PPP branch activity and hence supplying R5P for DNA synthesis and cell cycle progression.

--

Reviewer #3

<Please see attached PDF of the review>

--

Reviewer #4 (Remarks to the Author):

The work by Li et al proposes that the stability of transketolase like protein, TKTL1, is regulated by ubiquitylation and degradation during the cell cycle. The authors suggested that TKTL1 is stable at late G1 and S phase, when nucleotide biosynthesis are required, and it forms heterodimers with transketolase (TKT). This leads to the revers activity of the non-oxidative pentose phosphate pathway (PPP) and with that to the diversion of metabolites from glycolysis to ribose 5-phosphate (R5P) production. In addition, the authors claimed that TKT/TKTL1 heterodimer prevents the forward activity of the non-oxidative PPP, and hence prevents the diversion of R5P back into glycolysis. The authors further suggested that TKTL1 stability is controlled by the ubiquitin ligase action of the anaphase-promoting complex (APC) and its adaptor, CDH1. Increased nucleotide biosynthesis at S phase was suggested in the past and was recently nicely demonstrated by a temporal fluxomic approach (Molecular Systems Biology (2017) 13, 953). Furthermore, the role of TKTL1 in the non-oxidative PPP and nucleotide biosynthesis throughout the cell cycle was also suggested in the past (e.g. Tumour Biol. (2015) 36: 8519). However, the transketolase activity of TKTL1 was disputed over the years (e.g. BBA (2013) 1832: 387). The current work provided a potential explanation for this confusion. The authors suggest that indeed nucleotide metabolism is supported by TKTL1 during S phase and this is indeed mediated via the induction of the non-oxidative PPP. However, this activity of TKTL1 requires the interaction with TKT. As such, this is an interesting addition to this tale. The authors provided some evidence to support their claim, and addressed it in multiple ways. However, some major concerns still remained and some inconsistencies must be addressed.

Specific Comments:

1. In Figure 2 there are several problems: While a significant decrease in CDH1 could be achieved by siRNA (2j), the effect on TKTL1 ubiquitylation was minimal (2i) and no real effect on the endogenous levels of TKTL1 could be seen by western blot (Figure 2j; left), despite the presented bar graph (Figure 2j; right). Furthermore, the deletion of the D-Box in TKTL1 did not have a dramatic effect on protein ubiquitylation (Figure 2M). Overall, this questions the validity of the proposed model of TKTL1 regulation by APC.
2. The authors determined the Kd of TKT and TKTL1 by SPR. Here many technical details and quality assessments are missing. The purity of the proteins must be demonstrated, as it is not the enzymatic activity that is measured, but rather protein-protein interactions, and the SPR signal can be affected by many impurities. Further, the kinetics of the association and dissociation of the ligand to the immobilized protein must be shown (Figure 3D is meaningless as is).
3. The input levels and stoichiometry of TKT and TKTL1 in the size exclusion chromatography (Figure 3E) must be shown. It is very surprising that a complete shift of TKTL1 into a heterodimeric form was achieved with this assay.
4. The [1,2]-13C-glucose tracing is a key experiment in this study. However, there are a few observations made with this technique that remained problematic: The major labeled form of R5P in all conditions was the M2 form. In fact, the M1 form was almost negligible when the contribution of naturally-abundant 13C is considered. This demonstrate that under all conditions studied here, no major contribution of the oxidative PPP is seen. This is not in line with a score of glucose tracing experiments done in many cells (and in vivo) by many researcher, and to my opinion, it questions the technical validity of the experiments. In addition, if TKTL1 blocks the conversion of R5P to S7P (as suggested from the in vitro studies and the decrease in the M1 form of S7P - Figure 5G), one would expect to observe in Figure 5G an increase in the M1 form of R5P in TKTL1 overexpressed cells.
5. In Figure 5H the efficiency of TKT silencing was never shown. Furthermore, the effect of TKT silencing on R5P levels and its labeling pattern was not shown either since the results in Figures 5G-J are normalized to % change in comparison to GFP-transfected cells. The comparison in R5P between Figures 5G to 5H and between Figures 5I to 5J must be shown.

Point-to-point response to the reviewers' comments

Li, et al., “**APC/C^{CDH1} Synchronizes Ribose-5-Phosphate Levels and DNA Synthesis to Cell Cycle Progression**”. Manuscript ID: NCOMMS-18-12192.

Reviewer #1 (Remarks to the Author):

This manuscript by Li et al. et al. suggest that R5P is regulated in a cell cycle-specific manner by APC-mediated proteolytic turnover of TKTL1. Overall, the cell cycle fluctuation of TKTL1 and its corresponding impact on R5P levels are convincing. The data are also strong in supporting a role for ubiquitin-mediated proteolysis in generating TKTL1 periodicity. It is also clear that TKT levels impact R5P levels, probably through the TKTL1-TKT heterodimer. However, the authors' proposed model that TKTL1 levels are regulated by the APC/CCDH1 is not fully supported by the data. Many experiments presented show a consistent negative correlation between CDH1 levels and TKTL1 levels, but the evidence of a direct causal relationship is lacking. Since altering CDH1 levels could have numerous indirect effects on TKTL1, it is not possible to tell whether CDH1 directly regulates TKTL1 levels, and more experiments need to be done to thoroughly test this hypothesis. The authors write, “exogenous TKTL1 was found to interact with the SCF adaptors SKP2 and WD-40 domain protein 7 (FBXW7),” but they do not pursue this further since it did not repeat in the clear cell renal cell carcinoma (ccRCC) lines that they use throughout the paper. The potential involvement of the SCF in regulating TKTL1 levels may be worth investigating further given the incompatibility of some data with the hypothesis that the APC targets TKTL1.

All that said, this is an interesting manuscript. While there have previously been reports of the APC and SCF in targeting metabolic enzymes, this analysis is quite extensive and the nature of the branched and convergent metabolic pathways made

this study much more difficult than those on RNR, for example. In the end, whether it is the APC or the SCF doesn't change this importance of the work, but we need to know for sure.

Response: We thank the reviewer for his/her comments and suggestions. To show a direct causal relationship between CDH1 and TKTL1 expression, we first successfully generated a TKTL1 D-box mutant (TKTL1^{ΔD-box}-knockin) HeLa cell line (sequencing results are provided in revised Fig. S5C) using a Crispr-based knock-in approach. In contrast to that CDH1 expression decreased endogenous TKTL1 in TKTL1 wild-type cells (Fig. 2G), it failed to decrease endogenous mutant TKTL1 levels (revised Fig. 2O). Moreover, after double thymidine-synchronized G₁/S phase or RO3306-synchronized G₂/M phase of TKTL1^{ΔD-box}-knockin HeLa cells were released, endogenous mutant TKTL1 levels were not altered throughout the cell cycle, indicating that the D-box mutant abrogates the regulatory role of CDH1 on TKTL1 (revised Fig. S5D and E). Furthermore, in the TKTL1 mutant cells, either overexpression or knockdown of CDH1 did not affect R5P levels (revised Fig. S6B and C) and cell proliferation (revised Fig. S8E). These, together with findings that CDH1 interacted with and degraded TKTL1 but not the D-box mutant TKTL1 (Fig. 2L–N) and that the *in vitro* ubiquitination assay (Fig. 2K), confirmed that CDH1 directly regulates TKTL1 levels.

Regarding the potential involvement of SCF in regulating TKTL1 levels, we had examined the possibility and found, first, TKTL1 did not interact with the SCF adaptors SKP2 and FBW7 in tissue samples (Fig. S4D); second, overexpression of either SKP2 or FBW7 did not decrease cellular TKTL1 levels (Fig. S4E). These results suggest that SCF is not involved in TKTL1 regulation. These results were described in the last paragraph of the revised Result subsection “APC/C^{CDH1} mediates TKTL1 proteasomal degradation” (page 11, paragraph 2).

Revised Fig. S5C

Revised Fig. 20

Revised Fig. S5D

Revised Fig. S5E

Revised Fig. S6B

Revised Fig. S6C

Revised Fig. S8E

Specific problems with the APC data.

1. In figure 2H, overexpression of CDH1 appears to have no effect on TKTL1 levels. Similarly, knockdown of CDH1 in panel I of the same figure does not lead to an increase in TKTL1 levels. This is in contrast with later panels (e.g. figure 2J and 2N) where altering CDH1 levels appears to have a subtle effect on TKTL1 levels, and which the authors use as evidence that CDH1 directly regulates TKTL1. Similarly, the cycling of TKTL1 in panel 1B is not seen in figure 2O. The fact that data shows the desired result in the panels in which the authors wish to make a point, but not in the subsequent panels, is troubling.

Response: We thank the reviewer for their careful evaluation of our data. In Fig. 2H, we wanted to see the difference in ubiquitination levels of TKTL1 treated with or without CDH1 overexpression, and we normalized TKTL1 immunoprecipitated from cells of different treatments before resolving them on SDS PAGE. That's why TKTL1 levels were even. Similarly, for the revised Fig. 2P (original Fig. 2O), we normalized TKTL1 immunoprecipitated from different cell cycle to the same level and then compared the relative ubiquitination levels of immunoprecipitated TKTL1.

Original Fig. 2H

Revised Fig. 2P (original Fig. 2O)

2. The *in vitro* ubiquitination pattern of TKTL1 in figure 2K is very unusual.

Response: We optimized conditions and repeated this *in vitro* ubiquitination assay. The resulted ubiquitination pattern of TKTL1 looked better than previous one. We provided new result in revised Fig. 2K.

Revised Fig. 2K

Original Fig. 2K

3. In figure 2L the authors claim that the difference in binding between CDH1 and wild-type versus D-box mutant TKTL1 shows that CDH1 binds TKTL1 through its

D-box. However, the binding difference between wild-type and mutant is subtle, and, more importantly, the input level of the D-box mutant is actually less than the wild-type. Therefore, it is not clear that there is a loss of binding in the D-box mutant. Since this is a crucial experiment for their conclusion that CDH1 regulates TKTL1 levels, this is concerning. While the quantification of the data in Figure 2J appears to show a difference, this is not consistent with the gel that is shown.

Response: We thank the reviewer for their careful assessment of our results. Our previous data showed that similar amount of wild type TKTL1 bound more CDH1 than TKTL1 with D-box deleted, and this this was consistent with our conclusion that wild type TKTL1 binds CDH1 stronger than D-box mutant. To avoid the confusion of different input levels, we repeated the experiments when similar levels of TKTL1 and D-box mutant were expressed. As can be seen in the revised Fig. 2L, the binding affinity of CDH1 to D-box mutant TKTL1 was significantly lower than wild-type TKTL1, confirmed a loss of binding to CDH1 of the D-box mutant TKTL1.

Moreover, we repeated the experiment in Fig. 2J using two siRNAs targeting different regions within CDH1. Both siRNAs treatments resulted in TKTL1 accumulation. We have provided these results in revised Fig. 2J.

Revised Fig. 2L

Original Fig. 2L

Revised Fig. 2J

Original Fig. 2J

4. Figure 2M is either mis-labelled or improperly performed since the combinations of tagged proteins do not make sense. As is, the experiment does not compare ubiquitination of wild-type TKTL1 to the D-box mutant. This is a crucial experiment for their conclusion that CDH1 regulates TKTL1 levels, so the problem with this figure is concerning. Why is there so much ubiquitination in the D-box mutant?

Response: We thank the reviewer for the question raised. We apologize for mislabeling Fig. 2M. The concern that the D-box mutant was extensively ubiquitinated was likely due to excessive ectopic overexpression of TKTL1 led more ubiquitination. This was confirmed by lowering TKTL1 D-box mutant expression led much less ubiquitination. We provided the new results in revised Fig. 2M.

Revised Fig. 2M

Original Fig. 2M

5. If TKTL1 levels are directly regulated by the APC/CCDH1, then many of the R5P measurements in figure 6 and cell growth assays in figure 7 should be done with a TKTL1 D-box mutant to more directly test this regulation. If TKTL1 is a substrate of the APC/CCDH1, then disrupting the binding between the substrate and the APC should result in altered R5P levels. Although the experiments in figure 6 with CDH1 knockdown and overexpression show a consistent negative correlation between CDH1 levels and TKTL1 activity (measured by R5P and other metabolites), these only indirectly show an association between TKTL1 activity and the cell cycle and do not prove that CDH1 directly regulates TKTL1. As shown by their own data and that of others, altering CDH1 levels can alter the ratio of cells in cell cycle stages, which they show alters TKTL1 levels. I know of many cases where overexpression of CDH1 will strongly alter turnover that is SCF-mediated by altering cyclin levels. Specifically, an experiment like that in 1B, 1C, 5G, and 6C should be completed on a CRISPR generated endogenous D box mutant cell line.

Response: We thank the reviewer for their excellent suggestion and comments. Following the reviewer's suggestions, we generated TKTL1 Δ D-box mutant HeLa cells (TKTL1 ^{Δ D-box}-knockin HeLa) using CRISPR and performed the suggested experiments using this endogenous TKTL1-mutant cell line. The performed experiments and results obtained were as follows:

(1) We investigated the influence of endogenous TKTL1 D-box mutant on R5P synthesis by comparing the total R5P and isotopically traced metabolites between TKTL1 wild-type and TKTL1 mutant cells. We found, in TKTL1 ^{Δ D-box}-knockin HeLa cells, baseline R5P levels and non-oxidative PPP-derived R5P were higher than in TKTL1 wild-type cell lines. We described these observations in the Result (page 16, first paragraph, lines 1-3) and added them in revised Fig. 5H.

Revised Fig. 5H

(2) As to Fig. 1B and 1C, we detected protein expression in cells under DTB or RO3306 treatments. We found that for DTB or RO3306-synchronized TKTL1-mutant cells, expression levels of CDH1 and CDC20 were not affected during cell cycle progression. TKTL1 levels were not altered among the different cell cycle phases. We have discussed these findings in the Results (page 11, first paragraph, lines 4-6) and included them in revised Fig. S5, D and E.

Revised Fig. S5D

Revised Fig. S5E

(3) Regarding the experiments in Fig. 6C and 6D, we also performed them in the TKTL1 mutant cell line. We found that both knockdown and overexpression of CDH1

did not alter R5P levels. R5P levels only changed during cell cycle progression (Fig. S6B and C; page 17, first paragraph, lines 14-17).

Revised Fig. S6B

Revised Fig. S6C

ADDITIONAL, LESS CRITICAL POINTS ON THE FIGURES.

6. In figure 1E, the flag-tagged proteins are not specifically identified, and neither the text nor the figure legend explicitly explain what the anti-Flag is looking at.

Response: We apologize for not specifying the tag of the transfected proteins. We have included this information in revised Fig. 1E.

Revised Fig. 1E

Original Fig. 1E

7. The binding domain identification experiments in figures 3G and 3H are not very informative since the authors need to delete significant domains of TKT and TKTL1 to see loss of binding between the two proteins. These deletions are large enough that the remaining fragment may no longer fold, so the loss of binding does not support the idea that the absent domains are directly responsible for binding. If they want to support the hypothesis that these termini are sufficient for binding, they need to do that and remove the central portion of the molecule and show that they still bind.

Response: We appreciate the excellent suggestion by the reviewer. To validate, we removed the central portion of TKTL1 and TKT, and constructed new clones for co-immunoprecipitation assays. We confirmed that the TKTL1 protein without the central portion was still able to interact with TKT (revised Fig. 3I) and that the TKT protein without the central portion was able to interact with TKTL1 (revised Fig. 3J). The relevant findings have also been included in the Results (page 13, first paragraph, lines 2-4).

Revised Fig. 3I

Revised Fig. 3J

8. For the transketolase assays in figure 4 and the M1/M2 assays in figure 5, it would be informative to include absolute enzymatic activities and R5P levels in addition to the relative values presented in the figure.

Response: We thank the reviewer for the helpful suggestion. We measured the

absolute enzymatic activities of TKT, TKTL1, and the TKT-TKTL1 heterodimer. We provided absolute enzymatic activities, instead of relative enzymatic activities, of TKT, TKTL1, TKT homodimer, and TKT-TKTL1 heterodimer in revised Fig. 4B and 4C. The methods for measuring absolute enzymatic activities have been included in the Method section “Transketolase activity assay” (pages 29-30).

As for the isotopically traced assays in Fig. 5G–J, we quantified the absolute concentrations of R5P and S7P and included these values instead of the previously reported relative R5P levels. The original Fig. 5G and H were combined in revised Fig. 5G. The results in original Fig. 5I and J were replaced by the revised Fig. 5I and J. We have also included the method description in the Method section “LC-MS/MS measurement” (page 33, the end of the paragraph 2).

Revised Fig. 4B

Original Fig. 4B

Revised Fig. 4C

Original Fig. 4C

Revised Fig. 5G

Original Fig. 5G

Original Fig. 5H

Revised Fig. 5I

Original Fig. 5I

Revised Fig. 5J**Original Fig. 5J**
9. In figure 5, the authors add increasing amounts of TKTL1 to cells and lysates to show that this depletes TKT, suggesting that the TKTL1-TKT heterodimer is present in vivo. However, the authors do not comment on whether the exogenous TKTL1 levels—particularly in figures 5A and 5D—are comparable to endogenous levels that would actually occur at some stage of the cell cycle.

Response: We thank the reviewer for their insightful comment. Indeed, whether the cellular TKTL1 level is high enough to decrease TKT is critical for physiologic significance of TKTL1. We have included the quantitative results of endogenous TKTL1 in Fig. 1B, 1C, and levels of the exogenous and endogenous TKTL1 in Fig. 5A, 5D. We compared the levels of ectopically expressed TKTL1 in Figures 5A, 5D with those detected in cell phases in Figures 1B, 1C and found them comparable, indicate that physiologic levels of TKTL1 is capable of lowering TKT levels. We indicated this in the revised text as “When TKTL1 was progressively overexpressed to a level comparable to that of G₁/S phase in HeLa cells (see Figure 1B), removal of ectopically expressed TKTL1 from the cell lysates by affinity purification dose-dependently

decreased their TKT levels (Figure 5A)” (page 14, paragraph 2, lines 3-6).

Revised Fig. 1B

Revised Fig. 1C

Revised Fig. 5A

Revised Fig. 5D

- Minor point: some of the graph labels in figure 7H are cut off.

Response: We apologize for the mistake and have corrected it.

- There are a number of grammatical errors and confusing sentences in the text.

Response: We apologize for the lack in clarity and have improved the language by employing the help of a language-editing service from Elsevier webshop (<https://webshop.elsevier.com/languageservices/languageediting/>).

- The description of the CHX assay “Inhibition of protein translation by cycloheximide (CHX) failed to prevent TKTL1 levels from decreasing over time (Figure 2A)” while true, is very odd.

Response: We have edited this sentence, “Treating HeLa cells with cycloheximide, a protein translation inhibitor, did not prevent the degradation of TKTL1”.

- The species is not capitalized. E coli not E. Coli (page 11).

Response: We apologize for this oversight. We have corrected the mistake.

--

Reviewer #2 (Remarks to the Author):

This is an interesting and novel study in which Li and colleagues report the control of ribose-5-phosphate (R5P) levels by the regulation of the transketolase-transketolase-1-like protein (TKTKTL1) heterodimer formation. When TKTL1 is high and forms this complex, the formation of the TKT-TKT homodimer is prevented hence avoiding R5P removal from the forward non-oxidative branch of the pentose-phosphate pathway (PPP). In contrast, when the TKT-TKTL1 heterodimer is formed it facilitates the supply of R5P from glycolytically-derived carbon atoms through the reverse non-oxidative branch of the PPP. They also report that this effect is regulated by controlling TKTL1 stability by APC/C-CDH1 through a TKTL1 D-box. In this way, APC/C-CDH1 synchronizes cell cycle progression with R5P-derived DNA synthesis. The work is elegantly performed and the results are convincing. However, the authors have not put their data in the context of the very well-known role of APC/C-CDH1 on the control of glycolysis (The bioenergetic and antioxidant status of neurons is controlled by continuous degradation of a key glycolytic enzyme by

APC/C-Cdh1. Herrero-Mendez A, Almeida A, Fernández E, Maestre C, Moncada S, Bolaños JP. Nat Cell Biol. 2009 Jun;11(6):747-52. doi: 10.1038/ncb1881. Epub 2009 May 17. PMID: 19448625). By the way, the authors should acknowledge this NCB 2009 paper in the last sentence of the first paragraph of the introduction (ref. 10) because it was the first to show the regulation of PFKFB3 (hence glycolysis and PPP activity) by APC/C-CDH1, instead the PNAS-2010 that was erroneously cited.

Response: We thank the reviewer for bringing this to our attention. We have included this reference in the Introduction (page 4, last sentence of the first paragraph).

In fact, PFKFB3 (which is stabilized when CDH1 is low; see Herrero-Mendez et al., 2009) is a limiting factor to provide the necessary G3P that links glycolysis with R5P through the TKTL1-TKT heterodimer. It is therefore surprising that this tight link between the role of APC/C-CDH1 in controlling glycolysis has not been integrated in the message of this study. By doing so, this work could greatly enhance the interest to a wider readership and might support the possibility that inhibition of early steps of glycolysis may help stopping cancer cell progression by prohibiting carbon atoms supply to provide R5P. Accordingly, it would be nice if the authors could provide any experimental evidence that PFKFB3, which his controlled by APC/C-CDH1 activity (Herrero-Mendez et al., 2009), is actually playing a key role in providing substrate availability for the reverse non-oxidative PPP branch activity and hence supplying R5P for DNA synthesis and cell cycle progression.

Response: We thank the reviewer for their excellent suggestion. Following his/her suggestions, we provided experimental evidence that PFKFB3 also plays a role in supporting non-oxidative PPP metabolism:

(1) We constructed PFKFB3 knockout cells and monitored the levels of total R5P and double ¹³C-labeled R5P. We found that PFKFB3 knockout resulted in decreased

R5P synthesis, especially R5P from non-oxidative PPP (revised Fig. 5K). We have included related description in the Results (page 16, paragraph 1, lines 13-16).

(2) Moreover, PFKFB3 knockout saturated the influence of CDH1 on non-oxidative PPP-derived M2 R5P (Fig. 5L). We have included this finding in the Results (page 16, paragraph 1, lines 16-17).

(3) EDU assays showed that PFKFB3 knockout decreased DNA synthesis levels and weakened the effect of CDH1 on DNA synthesis (Fig. S7B). We have included this finding in the Results (page 19, first paragraph).

These results strongly support the reviewer's prediction and indicate that PFKFB3 is important in TKTL1-mediated R5P synthesis.

Revised Fig. 5K

Revised Fig. 5L

Revised Fig. S7B

Reviewer #3 (Remarks to the Author):

Review on the article “APC/CCDH1 Synchronizes Ribose-5-Phosphate Levels and DNA Synthesis to Cell Cycle Progression” by Yang Li, Fu-Jiang Xu, YuanYuan Qu, Jia-Tao Li, Yan Lin, Peng-Cheng Lin, Wei Xu, Jian-Yuan Zhao, and ShiMin Zhao.

The present study has revealed groundbreaking findings and we strongly recommend their publication with minor changes.

One of the most important hallmarks of life is the ability of cells to duplicate. Without this building of new cells, no life is possible. Therefore control of this duplication process is extremely important. In case of insufficient control unwanted cell proliferation and cancer can be the result. Several important signaling factors controlling cell duplication have been identified, but limited knowledge exists about the downstream mechanisms executing cell duplication. Since ribose / deoxyribose represent one of the major components of DNA / RNA, it is obvious that ribose metabolism will be involved in the DNA and RNA duplication process. Therefore the activity of enzymes regulating the building of ribose-5-phosphate (R5P), the crucial building block for DNA and RNA synthesis, is of special interest. R5P is an intermediary metabolite of the pentose phosphate pathway (PPP), required for both de novo and salvage synthesis of nucleosides and, consequently, for doubling of DNA / RNA synthesis. R5P sufficiency in PPP is a determining factor of DNA synthesis. It was found that the majority of R5P incorporated into de novo and salvage purine synthesis in S-phase comes from non-oxidative PPP, but the regulation of this process is still unclear. The current paradigm is that R5P is “pulled in” to the nucleotide and DNA synthetic pathways from the PPP, and the overproduction of nucleotides is prevented by feedback inhibition.

The data presented in this study demonstrate that R5P levels are proactively regulated via the APC/CCDH1-TKTL1 pathway and not via the hypothesized “pulled in” effect. This finding is of utmost importance for the understanding of DNA and RNA duplication, one of the most important processes in cell biology. Furthermore this new finding will enable a better understanding of mechanisms leading to diseases like cancer thereby offering the opportunity for new and better treatments of these diseases. Increased activation of cell proliferation represents the core problem of cancer. The data of this study pave the way to new treatments of cancer, since the APC/CCDH1-TKTL1 pathway represents a novel and promising target for future cancer therapies. In addition, insufficient DNA / RNA duplication caused by an inhibition of APC/CCDH1-TKTL1 pathway is also of importance, since this results in insufficient cell renewal and premature aging. These data of this study are important for both effects – diseases with activated and with inhibited DNA / RNA duplication. Overexpression of TKTL1 has been identified in all types of cancer, a disease characterized by activated DNA / RNA duplication. In addition, downregulation of TKTL1 has been detected in diseases characterized by insufficient self-renewal /premature aging or infertility.

The study of Li et al. used sophisticated techniques and experiments to unravel the mechanisms of R5P regulation. The design of the experiments as well as the performance of the experiments are of very high quality which enabled the discovery that APC/CCDH1 synchronizes R5P levels and DNA synthesis to cell cycle progression.

The positive correlation between TKTL1 levels and R5P levels in cell cycle suggested that TKTL1 may be a positive regulator of R5P. To test this possibility, the authors determined the R5P levels in TKTL1-overexpressing and -knocked down cells.

TKTL1 mRNA levels did not fluctuate during cell cycle progression, excluding the possibility that TKTL1 levels are regulated at the transcriptional level. TKTL1 controls

the building of R5P at the protein level by interacting with TKT transketolase.

Transketolase enzymes are known to be active as homodimers. For the first time, the authors of this study demonstrate that transketolase enzymes can be active as heterodimers. This finding itself will change the textbooks of human biochemistry, biology and medicine.

Response: We thank the reviewer for their comments.

The presence of TKT-TKTL1 heterodimers as well as its role in the regulation of transketolase activity has already been postulated by Coy et al., 1996 and should be mentioned. At that time TKTL1 was called TKR = transketolase related protein. This study is already cited by Li et al.

Extract of the discussion of Coy et al., 1996.

the brain-specific protein isoform. The C-terminus, which is the major part of the subunit interface in the transketolase homodimers, is identical in both the heart and the brain protein isoform of TKR. This raises the possibility that TKR and especially the heart-specific protein isoform of TKR can form active heterodimers or play a role in the regulation of transketolase activity. If TKR is the result of a gene duplication as

Response: We thank the reviewer for bringing this to our attention. We have discussed this and referenced it in the Results as “supporting previous ratiocination that TKTL1 and TKT can form active heterodimers and play a role in the regulation of transketolase activity” (page 12, paragraph 2, lines 9-11).

TKT homodimers are able to perform enzymatic reactions leading to a production of R5P (e.g. reaction 3), but also enzymatic reactions leading to a removal of R5P (e.g. reaction 1). Overexpression of TKTL1 leads to the building of heterodimers with

diminished reaction 1 activities, but elevated reaction 3 activities. Therefore transketolase heterodimers consisting of TKTL1 and TKT harbor significant changes in enzymatic activity leading to a shift from R5P removal to R5P production. The asymmetric transketolase activities of TKTL1-TKT heterodimer showed that the R5P removal (reaction 1) is impaired, but the R5P producing from non-oxidative PPP (reaction 3) is increased when the TKTL1-TKT heterodimer is formed in cells.

This finding is of utmost importance for the understanding of basic cellular mechanisms and for the understanding of diseases related to changes in cell division/proliferation. The data presented in this study represent an extremely important missing link between signaling of cell cycle and enzymatic processes executing cell cycle. A nearly 4-fold overexpression of TKTL1 leads to a 100% increase of R5P and a cellular situation in favor of cell duplication / proliferation. By determining the relative TKTL1 levels, a 4-fold overexpression of TKTL1 in cancer tissues (ccRCC tissues) as compared to corresponding adjacent non-cancer tissues could be identified concomitant with nearly doubled R5P levels in ccRCC tissues as compared to corresponding adjacent non-cancer tissues. These results confirm that cell cycle dependent variations in TKTL1 levels mediate cellular R5P levels. Furthermore these data underline the clinical relevance of TKTL1 overexpression and the concomitant increase of R5P favoring cell proliferation of cancer cells. Moreover, these data also explain why overexpression of TKTL1 is a prognostic marker for poor survival of cancer patients.

Response: We thank the reviewer for their comments.

However, we recommend making the following changes to the manuscript before publication in order to place the results in a correct overall context.

The authors write:

“In fact, TKTL1 is overexpressed in various cancers and is correlated with poor prognosis in colon and urothelial cancers 29-34.”

Besides colon and urothelial cancers, TKTL1 overexpression is a marker for poor prognosis in gastric, ocular adnexa, lung cancer, rectum carcinomas and laryngeal squamous cell carcinomas. This should be amended.

- TKTL1 and p63 are biomarkers for the poor prognosis of gastric cancer patients. Song Y, Liu D, He G. *Cancer Biomark.* 2015; 15(5):591-7. doi: 10.3233/CBM-150499.

- Enhanced TKTL1 expression in malignant tumors of the ocular adnexa predicts clinical outcome. Lange CA, Tisch-Rottensteiner J, Böhringer D, Martin G, Schwartzkopf J, Auw-Haedrich C. *Ophthalmology.* 2012 Sep; 119(9):1924-9. doi: 10.1016/j.ophtha.2012.03.037. Epub 2012 Jun 1.

- Poor outcome in primary non-small cell lung cancers is predicted by transketolase TKTL1 expression. Kayser G, Siene W, Kubitz B, Mattern D, Stickeler E, Passlick B, Werner M, Zur Hausen A. *Pathology.* 2011 Dec; 43(7):719-24. doi: 10.1097/PAT.0b013e32834c352b.

- Expression of Transketolase like gene 1 (TKTL1) predicts disease-free survival in patients with locally advanced rectal cancer receiving neoadjuvant chemoradiotherapy. Schwaab J, Horisberger K, Ströbel P, Bohn B, Gencer D, Kähler G, Kienle P, Post S, Wenz F, Hofmann WK, Hofheinz RD, Erben P. *BMC Cancer.* 2011 Aug 19; 11:363. doi: 10.1186/1471-2407-11-363.

- Overexpression of transketolase TKTL1 is associated with shorter survival in laryngeal squamous cell carcinomas. Völker HU, Scheich M, Schmausser B, Kämmerer U, Eck M. *Eur Arch Otorhinolaryngol.* 2007 Dec; 264(12):1431-6. Epub 2007 Jul 18.

Furthermore TKTL1 overexpression correlates with esophageal squamous cell

carcinoma and nasopharyngeal carcinoma metastasis.

- TKTL1 promotes cell proliferation and metastasis in esophageal squamous cell carcinoma. Li J, Zhu SC, Li SG, Zhao Y, Xu JR, Song CY. *Biomed Pharmacother.* 2015 Aug; 74:71-6. doi: 10.1016/j.biopha.2015.07.004. Epub 2015 Aug 3.
- [Effects of transketolase-like gene TKTL1 on occurrence and metastasis of human nasopharyngeal carcinoma]. Zhang S, Yang JH, Cai PC. *Zhonghua Yi Xue Za Zhi.* 2008 Dec 2; 88(44):3131-4. Chinese.

TKTL1 increase resistance against cisplatin chemotherapy.

- Knockdown of TKTL1 additively complements cisplatin-induced cytotoxicity in nasopharyngeal carcinoma cells by regulating the levels of NADPH and ribose-5-phosphate. Dong Y, Wang M. *Biomed Pharmacother.* 2017 Jan; 85:672-678. doi: 10.1016/j.biopha.2016.11.078. Epub 2016 Dec 6.

These studies should be included into the manuscript demonstrating both the clinical impact of TKTL1 overexpression and the APC/CCDH1-TKTL1 pathway for cancer patient survival and treatment.

Response: We greatly appreciate the reviewer's effort in providing us with so many useful studies regarding the relationship between TKTL1 and cancer. We have modified the sentence in the Introduction as "Indeed, TKTL1 is overexpressed in various cancers and is correlated with poor prognosis in colon, urothelial, gastric, and lung cancers as well as in ocular adnexa carcinomas, rectum carcinomas, and laryngeal squamous cell carcinomas. Increased TKTL1 levels also correlate with esophageal squamous cell carcinoma metastasis and increased resistance against cisplatin chemotherapy in nasopharyngeal carcinoma" (page 7, first paragraph, lines 7-12).

The authors should mention the study of Shi et al. demonstrating that downregulation of TKTL1 leads to inhibition of cell proliferation and cell cycle arrest in esophageal squamous cell carcinoma. This study strongly supports the findings of the study of Li et al. regarding the impact of the APC/CCDH1-TKTL1 pathway to cell cycle.

- TKTL1 expression and its downregulation is implicated in cell proliferation inhibition and cell cycle arrest in esophageal squamous cell carcinoma. Shi Z, Tang Y, Li K, Fan Q. *Tumour Biol.* 2015 Nov; 36(11):8519-29. doi: 10.1007/s13277-015-3608-7. Epub 2015 Jun 2.

Response: We thank the reviewer for their helpful suggestion. We have cited this reference in the last paragraph of the Introduction, “in contrast, TKTL1 downregulation attenuates the proliferation of various types of cancer cells” (page 7, first paragraph, lines 13-14).

The authors state: ‘Structural studies suggested that TKTL1 lacks transketolase activity,...’. In these structural studies a TKT protein harboring a deletion of 38 amino acids has been used for the analysis of TKTL1, but not TKTL1. The use of a deleted, artificial TKT protein is not suited for the analysis of TKTL1. In these studies no experiments with the TKTL1 protein have been done, demonstrating the lack of transketolase activity. Therefore Li et al. should mention that these structural studies have been performed with TKT and not with TKTL1, and only data generated with an artificially deleted TKT variant have been used to postulate statements regarding TKTL1 enzymatic activity.

Response: We have amended this description in the Introduction to “Compared with TKT, the most dissimilar region of TKTL1 was a 38-amino acid deletion in the C-terminus. Structural studies found that TKT proteins harboring this deletion lack transketolase activity, suggesting that TKTL1 lacks transketolase activity” (page 7,

first paragraph, lines 1-4).

The authors state: “The human genome encodes two proteins closely related to TKT: transketolase-like proteins 1 and 2 (TKTL1 and TKTL2) 25. To date, the physiological roles of TKTL1 and TKTL2 remain unknown.” In contrast to TKTL2, several important studies characterizing the physiological roles of TKTL1 have been published.

Response: We thank the reviewer for bringing this to our attention. We have deleted the following sentence, “To date, the physiological roles of TKTL1 and TKTL2 remain unknown.”

The authors state: “However, the transketolase activity of TKTL1 is yet to be confirmed.” Overexpression and downregulation of TKTL1 in cell lines clearly demonstrated the causal relationship between expression of TKTL1 and transketolase enzyme reaction in vivo. The following three studies harbor such experiments. The studies and these results should be integrated in the manuscript.

- TKTL1 is activated by promoter hypomethylation and contributes to head and neck squamous cell carcinoma carcinogenesis through increased aerobic glycolysis and HIF1alpha stabilization. Sun W, Liu Y, Glazer CA, Shao C, Bhan S, Demokan S, Zhao M, Rudek MA, Ha PK, Califano JA. *Clin Cancer Res*. 2010 Feb 1; 16(3):857-66. doi: 10.1158/1078-0432.CCR-09-2604. Epub 2010 Jan 26.
- Imatinib resistance associated with BCR-ABL upregulation is dependent on HIF-1alpha-induced metabolic reprogramming. Zhao F, Mancuso A, Bui TV, Tong X, Gruber JJ, Swider CR, Sanchez PV, Lum JJ, Sayed N, Melo JV, Perl AE, Carroll M, Tuttle SW, Thompson CB. *Oncogene*. 2010 May 20; 29(20):2962-72. doi: 10.1038/onc.2010.67. Epub 2010 Mar 15.

- A key role for transketolase-like 1 in tumor metabolic reprogramming. Diaz-Moralli S, Aguilar E, Marin S, Coy JF, Dewerchin M, Antoniewicz MR, Meca-Cortés O, Notebaert L, Ghesquière B, Eelen G, Thomson TM, Carmeliet P, Cascante M. *Oncotarget*. 2016 Aug 9; 7(32):51875-51897. doi: 10.18632/oncotarget.10429.

Response: We have included this information in the Introduction, "...is yet to be confirmed, especially through *in vitro* assays, although a correlation between TKTL1 and total cellular transketolase enzymatic activity was observed in cells" and included these references (page 6, last sentence; page 7, first sentence).

The study of Sun et al. 2010 could demonstrate that overexpression of TKTL1 in HNSCC cells promotes cellular proliferation and enhanced tumor growth *in vitro* and *in vivo*.

- TKTL1 is activated by promoter hypomethylation and contributes to head and neck squamous cell carcinoma carcinogenesis through increased aerobic glycolysis and HIF1alpha stabilization. Sun W, Liu Y, Glazer CA, Shao C, Bhan S, Demokan S, Zhao M, Rudek MA, Ha PK, Califano JA. *Clin Cancer Res*. 2010 Feb 1; 16(3):857-66. doi: 10.1158/1078-0432.CCR-09-2604. Epub 2010 Jan 26.

Response: We have changed the sentence in the Introduction to "Moreover, TKTL1 overexpression promotes cell proliferation and enhanced tumor growth; in contrast, TKTL1 downregulation attenuates the proliferation of various types of cancer cells" (page 7, first paragraph, lines 12-14).

Important changes in metabolism have been detected as a consequence of TKTL1 expression. Overexpression of TKTL1 increased the production of fructose-6-phosphate and glyceraldehyde-3-phosphate, in turn elevating the

production of pyruvate and lactate, resulting in the normoxic stabilization of the malignancy promoting transcription factor HIF1 α and the upregulation of downstream glycolytic enzymes.

The study of Zhao et al. 2010 showed that transketolase enzyme activity leads to resistance to imatinib (Gleevec) and transketolase enzyme inhibition by the thiamine analogue oxythiamine inhibited transketolase enzyme activity and re-sensitized cancer cells to imatinib.

- Imatinib resistance associated with BCR-ABL upregulation is dependent on HIF-1 α -induced metabolic reprogramming. Zhao F, Mancuso A, Bui TV, Tong X, Gruber JJ, Swider CR, Sanchez PV, Lum JJ, Sayed N, Melo JV, Perl AE, Carroll M, Tuttle SW, Thompson CB. *Oncogene*. 2010 May 20; 29(20):2962-72. doi: 10.1038/onc.2010.67. Epub 2010 Mar 15.

In this study important results regarding the enzymatic activity of TKTL1 have been observed: Proof of the enzymatic activity of TKTL1 has been shown, since stable expression of human TKTL1 protein in imatinib-resistant CML cells has been sufficient to prevent the effect of inhibition of TKT protein expression. This indicates that recombinant TKTL1 protein expression in cells (in vivo) is able to functionally complement the absence of TKT enzymatic activity leading to resistance to chemotherapy.

Response: We have cited this reference (no.28) in the Introduction to support the notion that TKTL1 may have transketolase activity (page 7, first sentence).

In 2016 a study about the enzymatic activity of TKTL1 and its physiological role has been published, which is very important for the understanding of the results observed by Li et al.

- A key role for transketolase-like 1 in tumor metabolic reprogramming. Diaz-Moralli S, Aguilar E, Marin S, Coy JF, Dewerchin M, Antoniewicz MR, Meca-Cortés O, Notebaert L, Ghesquière B, Eelen G, Thomson TM, Carmeliet P, Cascante M. *Oncotarget*. 2016 Aug 9; 7(32):51875-51897. doi: 10.18632/oncotarget.10429.

The study of Diaz-Moralli et al. identified an altered enzymatic reaction modus of TKTL1 based on the increase of the so called one-substrate reaction. The major enzymatic reaction of TKT is a two-substrate reaction and in contrast to TKT, TKTL1 is able to perform a one-substrate reaction leading to acetyl-CoA, an energy rich metabolite, playing a central role in anabolic processes. This energy rich metabolite is used for the building of lipids and other important building blocks for cell duplication. Furthermore TKTL1 represents the enzymatic basis for the so called Warburg effect – an oxygen independent glucose metabolism enabling a mitochondria independent ATP creation even in the presence of oxygen. The data presented in the study of Li et al. represent an important missing link to understand the whole picture. The data of Li et al. and Diaz-Moralli et al. complement each other and lead to a new picture, how cells duplicate. APC/CCDH1-TKTL1 pathway allows a metabolic switch enabling the production of R5P favoring duplication of cells. Furthermore due to the altered enzymatic activity of TKTL1 and its one-substrate reaction, acetyl-CoA as an energy rich building block for lipid and amino acid synthesis is generated. In addition TKTL1 allows an oxygen independent energy release supplying the cell with sufficient energy for cell duplication.

At that same time mitochondria are switched off, because the side effect of mitochondria based energy release is the production of reactive oxygen species (ROS). The switch from a mitochondria based energy release to a glucose fermentation based energy release is extremely important for cells, because during cell cycle oxidative stress can be minimized (Brand and Hermfisse, 1997).

- Aerobic glycolysis by proliferating cells: a protective strategy against reactive

oxygen species. Brand KA, Hermfisse U. *FASEB J.* 1997 Apr; 11(5):388-95.

During S-phase the nuclear membrane is dissolved and as a consequence mitochondria are in the same compartment of a cell as the duplication of DNA happens. At the time of S-phase, there is only one compartment – the cytoplasm and the free DNA being duplicated, but not a second compartment of DNA surrounded by the nuclear membrane. Active mitochondria and the reactive oxygen species they produce would damage the DNA during S-phase due to the absence of the nuclear membrane during S-phase. To avoid this, TKTL1 is used as an ATP generating metabolism free from the production of reactive oxygen species in the presence of oxygen. Therefore TKTL1 and its altered enzymatic properties causes a metabolic switch enabling a highly efficient and safe duplication of DNA and other cellular components, because the pool of R5P is increased favoring the duplication of DNA / RNA, but also the energy release is switched from mitochondria- to a TKTL1-based ATP generation avoiding the production of reactive oxygen species during cell cycle. Furthermore the acetyl-CoA, which is an extremely important building block for lipids and amino acids, is generated by TKTL1.(use this description in the discussion, may after showing the TKTL1 role in PPP regulation, cite another oncotarget paper as the reviewer required) Therefore the combination of increased R5P production concomitant with reactive oxygen free ATP production and creation of acetyl-CoA favors efficient and safe cell duplication during cell cycle. Due to this the importance of the APC/CCDH1-TKTL1 pathway is extremely high. The authors should integrate the results of the study of Diaz-Moralli in the discussion, since these data complement the important findings of Li et al.

Response: We greatly appreciate this helpful suggestion. We have integrated Diaz-Moralli's conclusions in the Discussion (page 22, last line; page 23, first paragraph, lines 1-2).

Downregulation or overexpression of TKTL1 causes several diseases and underline the clinical impact of the APC/CCDH1-TKTL1 pathway.

The in vivo effect of inhibition of TKTL1 protein in animals demonstrated the important role of TKTL1 for the prevention of radicals and cell damages. Lack of transketolase-like (TKTL) 1 aggravates murine experimental colitis.

- Lack of transketolase-like (TKTL) 1 aggravates murine experimental colitis. Bentz S, Pesch T, Wolfram L, de Vallière C, Leucht K, Fried M, Coy JF, Hausmann M, Rogler G. *Am J Physiol Gastrointest Liver Physiol*. 2011 Apr;300(4):G598-607. doi: 10.1152/ajpgi.00323.2010. Epub 2011 Jan 13.

Downregulation of TKTL1 due to mutations in the Werner-syndrome gene leads to high levels of reactive oxygen species and concomitant to high levels of DNA mutations. As a consequence patients with this gene mutation suffer from premature aging and death.

- Downregulation of the Werner syndrome protein induces a metabolic shift that compromises redox homeostasis and limits proliferation of cancer cells. Li B, Iglesias-Pedraz JM, Chen LY, Yin F, Cadenas E, Reddy S, Comai L. *Aging Cell*. 2014 Apr;13(2):367-78.

Furthermore a cell cycle with low levels of DNA mutations is extremely important for the generation of germ cells. As a consequence TKTL1 expression has been identified in germ cells and it could be demonstrated that TKTL1 could distinguish between semen from fertile and infertile men.

- Identification of genital tract markers in the human seminal plasma using an integrative genomics approach. Rolland AD, Lavigne R, Dauly C, Calvel P, Kervarrec C, Freour T, Evrard B, Rioux-Leclercq N, Auger J, Pineau C. *Hum Reprod*. 2013 Jan;28(1):199-209. doi: 10.1093/humrep/des360. Epub 2012 Sep 27.

The protective role of TKTL1 against radicals and cell damages also contributed to the evolution of cognitive functions of homo sapiens. TKTL1 belongs to a group of five genes, characterized by mutations specific for homo sapiens (compared to Neanderthals and apes), which were of utmost importance for the evolution of cognitive functions of modern humans.

- The complete genome sequence of a Neanderthal from the Altai Mountains. Prüfer K, Racimo F, Patterson N, Jay F, Sankararaman S, Sawyer S, Heinze A, Renaud G, Sudmant PH, de Filippo C, Li H, Mallick S, Dannemann M, Fu Q, Kircher M, Kuhlwilm M, Lachmann M, Meyer M, Ongyerth M, Siebauer M, Theunert C, Tandon A, Moorjani P, Pickrell J, Mullikin JC, Vohr SH, Green RE, Hellmann I, Johnson PL, Blanche H, Cann H, Kitzman JO, Shendure J, Eichler EE, Lein ES, Bakken TE, Golovanova LV, Doronichev VB, Shunkov MV, Derevianko AP, Viola B, Slatkin M, Reich D, Kelso J, Pääbo S. *Nature*. 2014 Jan 2;505(7481):43-9. doi: 10.1038/nature12886. Epub 2013 Dec 18.

The results of the study of Li et al. are of utmost importance for the understanding of a central mechanism of cell biology and life. The clinical impact of the APC/CCDH1-TKTL1 pathway and TKTL1 is overwhelming: malignancy of cancer, invasive growth and metastasis, resistance against chemotherapies and targeted therapies, but also premature aging, aggravation of ulcerative colitis, fertility and cognitive evolution of homo sapiens.

The results of the study of Li et al. unravel a novel biochemical pathway and its regulation by APC/CCDH1 with a strong impact on human health. This new knowledge will enable novel future therapies and preventive strategies for a broad range of diseases. The study of Li et al. represent a landmark study for biology and medicine.

I declare a conflict of interest since I am shareholder of Zyagnum AG (Pfungstadt,

Germany) and TAVARGENIX GmbH, (Pfungstadt, Germany) and owner of patents for the diagnostic and therapeutic use of TKTL1.

Best regards,

Dr. Johannes Coy 02.06.2018

Response: We thank the reviewer for providing so much information regarding the role of TKTL1 in disease. We greatly appreciate the reviewer's help and support.

--

Reviewer #4 (Remarks to the Author):

The work by Li et al proposes that the stability of transketolase like protein, TKTL1, is regulated by ubiquitylation and degradation during the cell cycle. The authors suggested that TKTL1 is stable at late G1 and S phase, when nucleotide biosynthesis are required, and it forms heterodimers with transketolase (TKT). This leads to the revers activity of the non-oxidative pentose phosphate pathway (PPP) and with that to the diversion of metabolites from glycolysis to ribose 5-phosphate (R5P) production. In addition, the authors claimed that TKT/TKTL1 heterodimer prevents the forward activity of the non-oxidative PPP, and hence prevents the diversion of R5P back into glycolysis. The authors further suggested that TKTL1 stability is controlled by the ubiquitin ligase action of the anaphase-promoting complex (APC) and its adaptor, CDH1. Increased nucleotide biosynthesis at S phase was suggested in the past and was recently nicely demonstrated by a temporal fluxomic approach (Molecular Systems Biology (2017) 13, 953). Furthermore, the role of TKTL1 in the non-oxidative PPP and nucleotide biosynthesis throughout the cell cycle was also suggested in the

past (e.g. *Tumour Biol.* (2015) 36: 8519). However, the transketolase activity of TKTL1 was disputed over the years (e.g. *BBA* (2013) 1832: 387). The current work provided a potential explanation for this confusion. The authors suggest that indeed nucleotide metabolism is supported by TKTL1 during S phase and this is indeed mediated via the induction of the non-oxidative PPP. However, this activity of TKTL1 requires the interaction with TKT. As such, this is an interesting addition to this tale. The authors provided some evidence to support their claim, and addressed it in multiple ways. However, some major concerns still remained and some inconsistencies must be addressed.

Specific Comments:

1. In Figure 2 there are several problems: While a significant decrease in CDH1 could be achieved by siRNA (2j), the effect on TKTL1 ubiquitylation was minimal (2i) and no real effect on the endogenous levels of TKTL1 could be seen by western blot (Figure 2j; left), despite the presented bar graph (Figure 2j; right). Furthermore, the deletion of the D-Box in TKTL1 did not have a dramatic effect on protein ubiquitylation (Figure 2M). Overall, this questions the validity of the proposed model of TKTL1 regulation by APC.

Response: We thank the reviewer for the insightful comments and apologize for changes not significantly enough in the figures. To address the reviewer's concern, we repeated these experiments again and provided the new results, which showed more pronounced changes, in revised Fig. 2I and 2J.

As for Figure 2M, the same question was raised by reviewer #1. The concern that the D-box mutant was extensively ubiquitinated was likely due to excessive ectopic overexpression of TKTL1 led more ubiquitination. This was confirmed by lowering TKTL1 D-box mutant expression led much less ubiquitination. We provided the new results in revised Fig. 2M.

Revised Fig. 2I

Original Fig. 2I

Revised Fig. 2J

Original Fig. 2J

Revised Fig. 2M

Original Fig. 2M

2. The authors determined the K_d of TKT and TKTL1 by SPR. Here many technical details and quality assessments are missing. The purity of the proteins must be demonstrated, as it is not the enzymatic activity that is measured, but rather protein-protein interactions, and the SPR signal can be affected by many impurities. Further, the kinetics of the association and dissociation of the ligand to the immobilized protein must be shown (Figure 3D is meaningless as is).

Response: We thank the reviewer for their helpful comments. First, we have supplemented the Coomassie brilliant blue (CBB) staining of both TKT and TKTL1 proteins in revised Fig. 3D. Second, we have provided detailed parameters for SPR in revised Fig. 3D, including the K_a (kinetics of the association), K_d (the dissociation constant), and K_D (the equilibrium dissociation constant).

Revised Fig. 3D

Binding affinity			
	k_a (1/Ms)	k_d (1/s)	K_D (M)
TKT—TKT	6.58×10^4	7.77×10^{-3}	1.18×10^{-7}
TKT—TKTL1	3.02×10^4	4.40×10^{-3}	1.46×10^{-7}

Original Fig. 3D

3. The input levels and stoichiometry of TKT and TKTL1 in the size exclusion chromatography (Figure 3E) must be shown. It is very surprising that a complete shift of TKTL1 into a heterodimeric form was achieved with this assay.

Response: We thank the reviewer for their suggestion. Following the reviewer's suggestion, we provided inputs information. The nearly complete shift of TKTL1 into a heterodimeric form was consistent with that TKTL1/TKT heterodimer is very stably formed between TKT and TKTL1 as we measured in revised Figure 3E.

Revised Fig. 3E

Original Fig. 3E

4. The [1,2]-¹³C-glucose tracing is a key experiment in this study. However, there are a few observations made with this technique that remained problematic: The major labeled form of R5P in all conditions was the M2 form. In fact, the M1 form was almost negligible when the contribution of naturally-abundant ¹³C is considered. This demonstrate that under all conditions studied here, no major contribution of the oxidative PPP is seen. This is not in line with a score of glucose tracing experiments done in many cells (and in vivo) by many researcher, and to my opinion, it questions the technical validity of the experiments. In addition, if TKTL1 blocks the conversion of R5P to S7P (as suggested from the in vitro studies and the decrease in the M1 form of S7P - Figure 5G), one would expect to observe in Figure 5G an increase in the M1 form of R5P in TKTL1 overexpressed cells.

Response: We thank the reviewer for the insightful comments. Oxidative PPP is constantly active in cells because it provides both R5P and NADPH. While some publications indicated oxidative PPP played important roles in maintaining cell growth,

lots of studies (references listed at the end of our response) indicated that the relative amount of R5P from oxidative PPP is lower than that from non-oxidative PPP in proliferating cells. Interestingly, the expression of oxidative PPP enzymes does not change during cell cycle progression; this indicates that oxidative PPP is not responsive to cell cycle progression (Figure 1B, 1C), providing further evidence that oxidative PPP unlikely provide regulation to R5P supply for DNA synthesis in S phase when R5P needs get higher. Together, we do agree with the reviewers that oxidative PPP contributes to R5P supply, however, based results from us from others, we think it is appropriate to conclude that R5P from non-oxidative PPP contributes more to DNA synthesis.

Regarding the results in Fig. 5G, although we observed an increase in M1 R5P in TKTL1-overexpressing cells, the difference was not significant. This is consistent with that M1 R5P from oxidative PPP making up a small portion of the overall R5P pool. Moreover, in proliferating such as TKTL1 overexpression cells, R5P was converted to PRPP for DNA synthesis. Therefore, neither M1 nor R5P should accumulate.

References:

- Fridman A, et al. Cell cycle regulation of purine synthesis by phosphoribosyl pyrophosphate and inorganic phosphate. *Biochem J.* 2013 Aug 15; 454(1):91-9. doi: 10.1042/BJ20130153.
- Ying H, et al. Oncogenic Kras maintains pancreatic tumors through regulation of anabolic glucose metabolism. *Cell.* 2012 Apr 27; 149(3):656-70. doi: 10.1016/j.cell.2012.01.058.
- Shukla SK, et al. MUC1 and HIF-1alpha Signaling Crosstalk Induces Anabolic Glucose Metabolism to Impart Gemcitabine Resistance to Pancreatic Cancer. *Cancer Cell.* 2017 Sep 11; 32(3):392. doi: 10.1016/j.ccell.2017.08.008.
- Boros LG, et al. Nonoxidative pentose phosphate pathways and their direct role

in ribose synthesis in tumors: is cancer a disease of cellular glucose metabolism?
 Med Hypotheses. 1998 Jan; 50(1):55-9.

- Cascante M, et al. Role of thiamin (vitamin B-1) and transketolase in tumor cell proliferation. Nutr Cancer. 2000; 36(2):150-4. Review.

5. In Figure 5H the efficiency of TKT silencing was never shown. Furthermore, the effect of TKT silencing on R5P levels and its labeling pattern was not shown either since the results in Figures 5G-J are normalized to % change in comparison to GFP-transfected cells. The comparison in R5P between Figures 5G to 5H and between Figures 5I to 5J must be shown.

Response: We thank the reviewer for bringing this to our attention. In the revised manuscript, we have included the efficiency of TKT knocking down in revised Fig. 5G. For comparing the labeling patterns of R5P between the mentioned figures, we have provided the absolute quantification results for the figures the reviewer mentioned.

Revised Fig. 5G

Original Fig. 5G

Original Fig. 5H

Revised Fig. 5I

Original Fig. 5I

Revised Fig. 5J

Original Fig. 5J

REVIEWERS' COMMENTS:

Reviewer #1 (Remarks to the Author):

Changes to the manuscript address most of my concerns. I support publication of the current version of the manuscript.

Reviewer #2 (Remarks to the Author):

This reviewer is very happy to see that the authors have now integrated the role of APC/C-CDH1 at regulating PFKFB3 stability (and, hence, glycolysis) with the herein described TKT-L1-mediated R5P synthesis. This highlights the impact of the finding that APC/C-CDH1-mediated regulation of glycolysis and PPP activity takes place by controlling the protein stability of PFKFB3 (Herrero-Mendez et al., 2009). The authors therefore have adequately addressed this reviewer's concern, although they have not removed the reference 10, as this reviewer requested, in the introductory paragraph, because it is incorrect to state that Almeida et al. (2019) identified the regulation of PFKFB3 by APC/C-CDH1. Please, amend. In addition, the authors have not discussed the link between APC/C-CDH1-mediated regulation of PFKFB3 and carbon atoms supply for R5P, as they have now shown in Fig. 5K,L, Fig. S7B. Adding a sentence integrating the direct control of glycolysis (Herrero-Mendez et al., 2009) and PPP (this work) by APC/C-CDH1 would incredibly rise the impact of the discussion, at the view of this reviewer.

Reviewer #4 (Remarks to the Author):

The authors have addressed my initial concerns and in my view, the manuscript is now publishable.

Point-to-point response to the reviewers' comments

Li, et al., “**APC/C^{CDH1} Synchronizes Ribose-5-Phosphate Levels and DNA Synthesis to Cell Cycle Progression**”. Manuscript ID: NCOMMS-18-12192A.

Reviewer #1 (Remarks to the Author):

Changes to the manuscript address most of my concerns. I support publication of the current version of the manuscript.

A: We thank the Reviewer for all his/her help.

Reviewer #2 (Remarks to the Author):

This reviewer is very happy to see that the authors have now integrated the role of APC/C-CDH1 at regulating PFKFB3 stability (and, hence, glycolysis) with the herein described TKT-L1-mediated R5P synthesis. This highlights the impact of the finding that APC/C-CDH1-mediated regulation of glycolysis and PPP activity takes place by controlling the protein stability of PFKFB3 (Herrero-Mendez et al., 2009). The authors therefore have adequately addressed this reviewer's concern, although they have not removed the reference 10, as this reviewer requested, in the introductory paragraph, because it is incorrect to state that Almeida et al. (2019) identified the regulation of PFKFB3 by APC/C-CDH1. Please, amend. In addition, the authors have not discussed the link between APC/C-CDH1-mediated regulation of PFKFB3 and carbon atoms supply for R5P, as they have now shown in Fig. 5K,L, Fig. S7B. Adding a sentence integrating the direct control of glycolysis (Herrero-Mendez et al., 2009) and PPP (this work) by APC/C-CDH1 would incredibly rise the impact of the discussion, at

the view of this reviewer.

A: We thank the Reviewer for the helpful suggestions. Following the reviewer's suggestion, we removed the reference 10 (Almeida et al. 2010) in the Introduction section. Moreover, we discussed the direct control of glycolysis and PPP by APC/C^{CDH1} in the Discussion as "Considering that APC/C^{CDH1} also controls the protein stability of PFKFB3, a rate-limiting enzyme of glycolysis,¹⁰ we validate PFKFB3 is a limiting factor to provide the necessary G3P that links glycolysis with R5P through non-oxidative PPP (see Figures 5K, L). These results indicate a direct control of PPP and glycolysis by APC/C^{CDH1} and suggest a possibility that activation of APC/C^{CDH1} may help inhibiting cancer cell progression by prohibiting R5P generation through the inhibition of PPP and early steps of glycolysis" (lines 3-9, paragraph 1, page 22).

Reviewer #4 (Remarks to the Author):

The authors have addressed my initial concerns and in my view, the manuscript is now publishable.

A: We thank the Reviewer for all his/her help.